# Molecular encoding and synaptic decoding of context during salt chemotaxis in *C. elegans*

Shingo Hiroki[1], Hikari Yoshitane[1,2], Hinako Mitsui[1], Hirofumi Sato[1], Chie Umatani [1], Shinji Kanda[3], Yoshitaka Fukada [1,2,4] & Yuichi Iino [1✉]

Animals navigate toward favorable locations using various environmental cues. However, the mechanism of how the goal information is encoded and decoded to generate migration toward the appropriate direction has not been clarified. Here, we describe the mechanism of migration towards a learned concentration of NaCl in *Caenorhabditis elegans*. In the salt-sensing neuron ASER, the difference between the experienced and currently perceived NaCl concentration is encoded as phosphorylation at Ser65 of UNC-64/Syntaxin 1 A through the protein kinase C(PKC-1) signaling pathway. The phosphorylation affects basal glutamate transmission from ASER, inducing the reversal of the postsynaptic response of reorientation-initiating neurons (i.e., from inhibitory to excitatory), guiding the animals toward the experienced concentration. This process, the decoding of the context, is achieved through the differential sensitivity of postsynaptic excitatory and inhibitory receptors. Our results reveal the mechanism of migration based on the synaptic plasticity that conceptually differs from the classical ones.

[1] Department of Biological Sciences, School of Science, The University of Tokyo, Tokyo, Japan. [2] Circadian Clock Project, Tokyo Metropolitan Institute of Medical Science, Tokyo, Japan. [3] Atmosphere and Ocean Research Institute, The University of Tokyo, Chiba, Japan. [4] Laboratory of Animal Resources Center for Disease Biology and Integrative Medicine, Graduate School of Medicine, The University of Tokyo, Tokyo, Japan. ✉email: iino@bs.s.u-tokyo.ac.jp

Navigation is one of the most fundamental and important behaviors that are widely conserved in various animals. Animals use various sensory cues to reach their favorable locations, such as food patches or nests. The direction of migration relative to the gradient of sensory cues during navigation is often plastic depending on the context such as physiological state, current position, or past experience. For instance, the use of visual landmark switches according to contexts, such as whether the honeybee is homing or foraging[1,2].

Especially when the intensity of sensory cues (i.e., temperature, concentration of substances, etc.) is associated with the goal, the behavioral response to the stimulus input must be dynamically regulated; when the input level currently perceived by the animal is higher than that expected at the goal, the animal should migrate in the direction in which the sensory input is reduced, and vice versa[3]. However, a comprehensive understanding of how the context, such as relative intensity of sensory cues compared to that expected at the goal, is encoded in the nervous system by specific molecules and how this molecular code is decoded later as a bidirectional behavioral response to sensory input after navigation onset has not been fully understood to date.

The nematode Caenorhabditis elegans exhibits navigation behavior in response to various sensory cues such as temperature (thermotaxis), volatile odorant (odor chemotaxis), and NaCl (NaCl chemotaxis), during which the direction of the migration is plastically regulated based on the context[4–6]. Intriguingly, in these navigation behaviors, the activity of protein kinase C, PKC-1, in each sensory neuron determines the direction of migration[4,7,8]. The inactivation of PKC-1 in the thermosensory neuron AFD causes thermophilic behavior, whereas the activation of PKC-1 causes cryophilic behavior regardless of contexts[8]. Likewise, the inactivation of PKC-1 in olfactory neuron AWC causes aversion of appetitive odor cues, whereas the activation of PKC-1 causes enhancement of attraction to the odor cues[7].

Interestingly, the activity of PKC-1 in the sensory neuron is dynamically regulated to guide the worms to preferred concentration of salt (NaCl). When C. elegans were cultivated with varied concentrations of NaCl for several hours and then placed onto an NaCl gradient, they migrate toward the experienced concentration (Fig. 1a, b)[4]. In this behavior, a single NaCl-sensing neuron called ASER is sufficient to drive navigation, and the activity of PKC-1 in ASER is a key factor that determines the direction of migration in this behavior[4]; when PKC-1 is active, the worms migrate toward higher NaCl concentrations, and the opposite is true when PKC-1 is inactive (Fig. 1b). Importantly, the amount of the PKC-1 activator, diacylglycerol (DAG), in ASER changes based on differences between the concentrations currently perceived on gradient plate and that previously experienced during conditioning (Fig. S1a, b)[9]. The change in DAG is maintained for tens of minutes in ASER, which coincides with the time window in which the direction of migration is regulated to drive the animal toward the cultivation concentration (Figs. 1e and S1a, b)[9]. Since DAG regulates chemotaxis via PKC-1 in standard conditions[10], DAG/PKC pathway is considered to play canonical roles in salt chemotaxis. However, as in other navigation behaviors, the mechanism downstream of PKC-1 remains to be unknown.

Here, we describe the mechanisms of how the context regarding NaCl concentrations is encoded into molecules through DAG/PKC pathway and how this change is decoded as bidirectional migration, based on the behavioral response to changes in NaCl concentration perceived on the concentration gradient. The difference between currently perceived and previously experienced NaCl concentration is expressed as the activity of PKC-1 and is encoded as the level of 'basal' glutamate release via phosphorylation at Ser65 of UNC-64/Syntaxin 1A. The change in basal release can be decoded as reversal of postsynaptic response using difference in sensitivity between excitatory and inhibitory glutamate receptors. Taken together, our results describe an encoding and decoding of contexts via the synaptic plasticity different from classical ones, which underlies a plastic switch of migration direction.

## Results

**Neuron-specific phosphoproteomic analysis revealed phosphorylation at Ser65 of UNC-64 downstream of PKC-1.** C. elegans navigates toward NaCl concentration experienced during previous cultivation. To quantify the migration bias according to context, we followed the previous report[4] with slight modifications. Briefly, animals were cultivated at three different concentrations (100, 50, and 25 mM) for ~6 h, then transferred to the center of the assay plate with NaCl gradient ranging from ~30 to ~90 mM, the center of which is 50 mM (Fig. 1a). As previously reported[4], while wild type N2 strain show context-dependent navigation behavior, pkc-1(nj3[W218stop], loss of function) mutation[8] caused strong migration bias toward lower NaCl concentration. This bias was rescued by expression of pkc-1 in ASER, and expression of the constitutively active form of pkc-1(A160E, gain of function)[8] in ASER resulted in migration bias toward higher NaCl concentration (Fig. 1b). When sensory perception of ASER, which is caused by change in NaCl concentration (50–25 mM), was observed using microfluidic chip and GCaMP2.1, no abnormality was seen in pkc-1(nj3lf) mutants or pkc-1(gf)-expressing transgenic animals (Fig. 1c, d).

We set out to explore the downstream molecules of PKC-1 to reveal the underlying mechanism in the encoding process. We used a phosphoproteomic approach for identification: we compared the phosphoproteomic profile of pkc-1(lf) mutants with that of worms that pan-neuronally express an active form of pkc-1(A160Egf). Since it is practically impossible to physically dissect the nervous system of worms due to their small size, we first extracted proteins from the whole body. However, the phosphoproteomic profile did not seem to accurately reflect neural events (Fig. S2a, b), because it had relatively small number of neuron-specific proteins and de-phosphorylation events were mainly captured in PKC-1-activated strains, contradicting an intuition. This is likely due to the small volume of neural tissue compared to others such as the intestine. Therefore, highly effective neuron-specific protein extraction was required. We labeled the neural proteins by a pan-neuronal expression of TurboID, a modified non-specific biotinylation enzyme[11] (Fig. 2a, b). We purified the biotin-labeled proteins by streptavidin beads, and analyzed them through a liquid chromatograph mass spectrometry. As a result, we obtained the phosphoproteomic dataset which contained larger fraction of phosphopeptides derived from neuron-enriched transcripts[12] (~32% in fraction) (Fig. 2c, Table 1; see also Supplementary Data S1 and S2). In this profile, the total number of phosphopeptides were increased by PKC-1 activation, consistent with our intuition. Moreover, the phosphopeptides derived from PKC-1 itself and the known nPKC target protein, DKF-2[13], were enriched in pkc-1(gf) dataset (Fig. 2d, e). However, it should be noted that known PKC motif, RxxSxR, was not enriched in this dataset, suggesting that a certain amount of indirect phosphorylation was still involved the dataset or PKC-1 has consensus motif(s) different from well-known PKCs such as mammalian PKCs.

Therefore, this phosphoproteomic dataset seems to capture the target proteins that function downstream of PKC-1 in neurons.

Given that PKC-1 does not affect the sensory activation of ASER itself (Fig. 1c, d), we searched for a phosphorylation of synaptic protein regulated by PKC-1, which would lead to

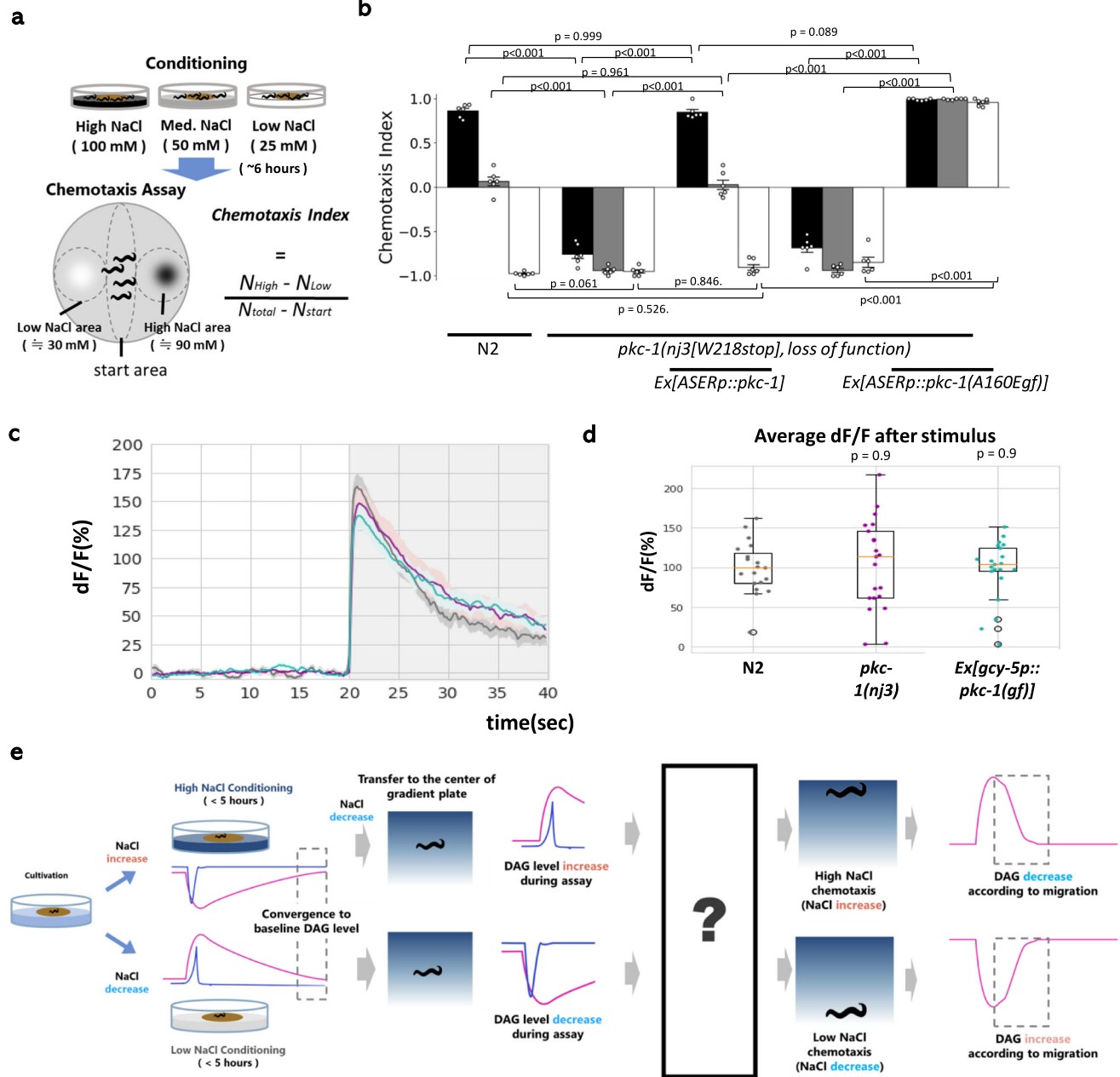

**Fig. 1 Activity of PKC-1 in NaCl-sensing neuron ASER controls the navigation toward experienced NaCl concentration. a** Schematic illustration of the NaCl concentration-learning paradigm[4]. *C. elegans* animals are incubated with food at one of the three concentrations of NaCl (100, 50, and 25 mM), and then transferred to an agar plate with gradient of salt concentrations at the start area with ~50 mM of NaCl and allowed to freely migrate. Chemotaxis Index was calculated as indicated based on the numbers of animals in each area at the end of the assay. **b** Wild-type animals show chemotaxis depending on the NaCl concentration during conditioning, while manipulation of *pkc-1* causes altered behaviors showing that PKC-1 functions in the gustatory sensory neuron ASER to determine the direction of migration on the NaCl gradient. $n = 6$ independent experimental repeats(assays). Error bars indicate SEM. n. s., $p > 0.05$, ***$p < 0.001$ in the Tukey's test (multiple comparisons). **c** Activation of ASER was observed using GCaMP2.1. The NaCl decrease (50–25 mM) is indicated with a gray shade. $n = 20$ independent animals. each color represents the strain: gray: N2, magenta: *pkc-1(nj3)*, green: *Ex[gcy-5p::pkc-1(gf)]*. **d** Quantification results of (**c**). *p* values were calculated by Tukey's test. The boxes extend from 25th to 75th percentile (first and third quartiles), median is marked by the line, and the ends of whiskers indicate the minimum and maximum values within 1.5× the inter-quartile range from the first and third quartiles, respectively[9]. **e** An illustration of the model of navigation regulated by DAG/PKC. When animals are transferred from growth plates to conditioning plates, following calcium influx or efflux in ASER (blue), the amount of DAG (pink) in ASER either increases or decreases, according to the difference in the concentration of NaCl between the growth plate and the conditioning plate. However, after ample conditioning time, the amount of DAG returns to baseline. When worms are transferred from the conditioning plate to the center of the NaCl gradient, DAG either increases or decreases depending on the difference between the NaCl concentration at the conditioning plate and the center of the gradient plate. Increased or decreased DAG regulates the activity of PKC-1, and PKC-1 signals to unknown molecules to regulate unknown neural processes. Source data are provided as a Source Data file.

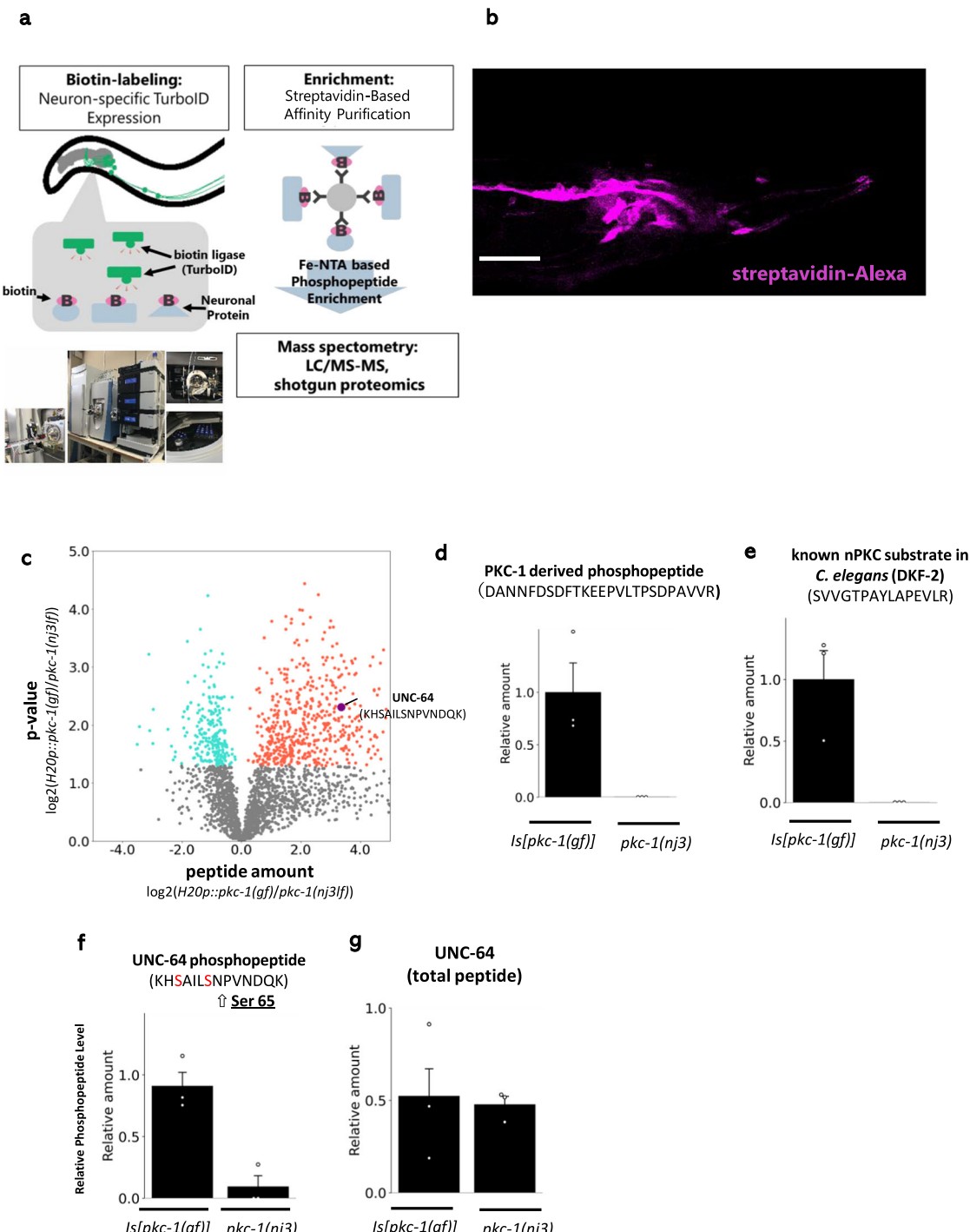

**Fig. 2 Neuro-phosphoproteomic analysis using non-specific biotinylation enzyme, TurboID. a** Schematic illustration of neuro-phosphoproteomic analysis using TurboID. **b** Biotin label in *rgef-1p*::*TurboID* was visualized using streptavidin-Alexa. The label appeared to be restricted in the nervous system. At least two animals observed by confocal microscopy showed similar pattern, with multiple animals appeared to be similarly stained under fluorescence microscopy. Scale bar: 40 μm. **c** Results of neuro-phosphoproteomic analysis. *pkc-1(nj3[W218stop], loss of function)* and pan-neuronal *H20p*::*pkc-1(A160E gain of function)*-expressing strains were compared. Volcano plot of phosphoproteomic analysis. Red and blue dots represent upregulated and downregulated ($P < 0.05$ in two-sided t-test) phosphopeptides, respectively, in *pkc-1(gf)* expressed strains, compared to *pkc-1(nj3lf)* mutants. Other phosphopeptides are indicated as gray dots. The purple dot represents the peptide KHSAILSNPVNDQK, which is mapped onto UNC-64. The average of the triplicates ($n = 3$ independent experiments) is described. Peptides which were not detected in either strains were not represented. **d** The amount of phosphopeptides derived from PKC-1. **e** The amount of phosphopeptides from a known PKC substrate, DKF-2. **f** The amount of phosphopeptides containing UNC-64 Ser65. **g** The amount of total peptides containing UNC-64. Error bars indicate SEM for all panels in Fig. 2. Source data are provided as Supplemental Data.

**Table 1 The fraction of neural genes included in the phosphoproteomic dataset.**

| Classification | Phospho-peptide count |
|---|---|
| Upregulated phosphopeptide ($P < 0.05$) | 526 |
| Downregulated phosphopeptide ($P < 0.05$) | 256 |
| Total phosphopeptide identified | 2555 |
| Fraction of peptide from neuron-specific genes | 32.74% |

Summary of phosphoproteomic analysis. The list of neuron-specific genes was obtained from Kaletsky et al., 2016.

changes in neurotransmitter release downstream of neuronal activation. Among the phosphorylation sites upregulated by PKC-1, we found the > 10 fold change in phosphopeptide levels containing UNC-64 Ser65 or Ser61. UNC-64 is a *C. elegans* homolog of Syntaxin 1A, which is a widely conserved t-SNARE that plays a critical role in neurotransmission[14,15]. Because there is no significant difference in total peptide amount derived from UNC-64 between *pkc-1(lf)* and *pkc-1(gf)*, the difference in phosphopeptides reflects the difference in phosphorylation events (Fig. 2f, g). These residues, Ser61 and Ser65, are phosphorylation sites well-conserved among a wide range of organisms (Fig. S3a–c), and are located in the Habc domain[16,17]. The Habc domain interacts with the H3 t-SNARE domain to form an inactive state of Syntaxin ('closed' form)[17]. Switch from active ('open') Syntaxin to inactive ('closed') Syntaxin is dynamically regulated by molecular chaperone proteins such as UNC-18/Munc18 or UNC-13/Munc13 to control neurotransmitter release[18].

**UNC-64 Ser65 phosphorylation affects a level of glutamate release from ASER to generate the context-dependent navigation.** To test whether UNC-64 Ser61 or Ser65 contributes to the migration on the NaCl gradient, we generated phospho-deficient mutants of these sites (*unc-64(S65A)* and *unc-64(S61A)*) using CRISPR/Cas9-based genome editing[19]. *unc-64* S65A substitution resulted in obvious migration bias toward low NaCl concentrations (Fig. 3a), a trend similar to that caused by *pkc-1(nj3lf)*. On the other hand, u*nc-64* S61A did not show any migration defect (Fig. S4b). Note that unc-64 S61E and S65E did not show any defect in chemotaxis, suggesting that glutamate substitution does not mimic phosphorylated state (Fig. S4a).

To test whether *unc-64* Ser65 functions in ASER, we inserted a single-copy transgene[20] that pan-neuronally expresses wild-type *unc-64*. The ORF of *unc-64* in this transgene is sandwiched by *frt*, thus can be excised by the recombinase FLP. The expression of transgene partially rescued the migration bias of *unc-64(S65A)*, whereas the expression of FLP under the ASER-specific promoter regenerated the bias (Fig. 3b), showing that UNC-64 Ser65 indeed functions in ASER.

Next, we examined which neurotransmitters are regulated by UNC-64 Ser65 and PKC-1. While one previous report demonstrated that PKC-1 modifies neuropeptidergic transmission[21], there is another study that demonstrated that PKC-1 regulates glutamatergic transmission in sensory neurons[22]. Because ASER expresses both glutamate and neuropeptides as neurotransmitters[23,24], we first tested whether glutamate transmission-deficient (*eat-4(ky5)*)[25] or dense core vesicle (neuropeptide) release-deficient (*unc-31(e928)*)[26] mutants suppress the migration bias toward high NaCl caused by ASERp::*pkc-1(gf)* (Fig. 3c). The single mutations could not suppress the effect of ASERp::*pkc-1(gf)* while double mutations did, showing that both glutamate and neuropeptide contribute to driving the

migration direction to higher NaCl concentrations under activated PKC-1. However, in contrast to ASERp::*pkc-1(gf)*, the effect of *unc-64(S65A)* was completely suppressed by ASER-specific *eat-4* knockout using FLP/FRT[27] (Fig. 3d). In this experiment, ASER-specific knockout of *eat-4* itself did not appear to cause a severe defect in navigation, suggesting the redundant role of neuropeptide in sensory transduction from ASER. These results suggest that the phosphorylation of UNC-64 Ser65 might specifically regulate glutamatergic transmission to generate chemotactic bias, while another pathway regulates neuropeptide release downstream of PKC-1 (Fig. 3e). To further test this possibility, we observed neuropeptide release using an established assay[21] in *unc-64(S65A)* (Fig. S5b, c). Neuropeptides released from motor neurons are uptaken by scavenger cells called coelomocytes in *C. elegans*. Therefore, when fluorescent proteins fused with neuropeptide are expressed in motor neurons, fluorescence in coelomocytes reflects release of neuropeptide-fused YFP. Because UNC-64 itself is necessary for neuropeptide release[26], a previous study showed coelomocyte fluorescence is reduced in *unc-64* loss of function mutants[21]. However, *unc-64(S65A)* did not cause significant reduction in coelomocyte fluorescence (Fig. S5c). Note that *unc-64(S65A)* shows lower aldicarb sensitivity (Fig. S5a), which indicates reduced acetylcholine neurotransmission in motor neurons[28]. As expected from this assay, *unc-64(S65A)* showed a small reduction in locomotion speed (Fig. 3f), but the effect is moderate and is unlikely to explain the altered chemotaxis. Thus, UNC-64 Ser65 might specifically regulate the release of synaptic vesicles but not neuropeptides.

To examine how the glutamate release from ASER change in *pkc-1(nj3lf)* and *unc-64(S65A)*, we observed glutamatergic transmission from ASER in the mutants using *eat-4/vGluT*-fused pHluorin (Fig. 4a)[22]. Decrease in NaCl concentration (50–25 mM) caused glutamate release from ASER, and both *pkc-1(nj3lf)* and *unc-64(S65A)* mutants exhibited reduced glutamatergic transmission (Fig. 4b, c).

Interestingly, the open syntaxin mutant (*unc-64(L166A/E167A)*)[29] only slightly affected the chemotaxis and did not suppress *pkc-1(nj3)* (Fig. S4c), suggesting that *unc-64(S65A)* might have functions other than merely shifting the switch between closed/open forms of the molecule.

Taken together, context, or experience, is encoded as altered levels of glutamatergic transmission from ASER by PKC-1 and UNC-64 Ser65, which in turn determines the migration bias to higher or lower NaCl concentrations.

**Change of glutamate transmission in *unc-64(S65A)* and *pkc-1(nj3)* reverses the postsynaptic response.** Next, we investigated how this presynaptic reduction is decoded as migration bias to low NaCl. Here we focused on the activity of the interneuron AIB. AIB is an interneuron that functions in reorientation behavior during migration via known backward locomotion circuit[30,31]. Importantly, AIB directly receives glutamatergic synaptic input from ASER[30], and controls the reorientation-based chemotactic behavior[4,30]. Furthermore, a recent report demonstrated that the response of AIB to the decrease or increase in NaCl depends on glutamatergic input from ASER, which can be either excitatory or inhibitory depending on the context, that is, the difference between NaCl concentration in the cultivation phase and that currently perceived in the imaging environment (Fig. 5a)[30]. Therefore, the mechanism which underlies plastic connection between ASER and AIB may reflect the fundamental basis of plastic navigation behaviors.

By using microfluidic arena[30,32], we observed the response of AIB in freely moving animals that were conditioning at 50 mM. In wild-type animals, AIB shows excitatory responses to concentration

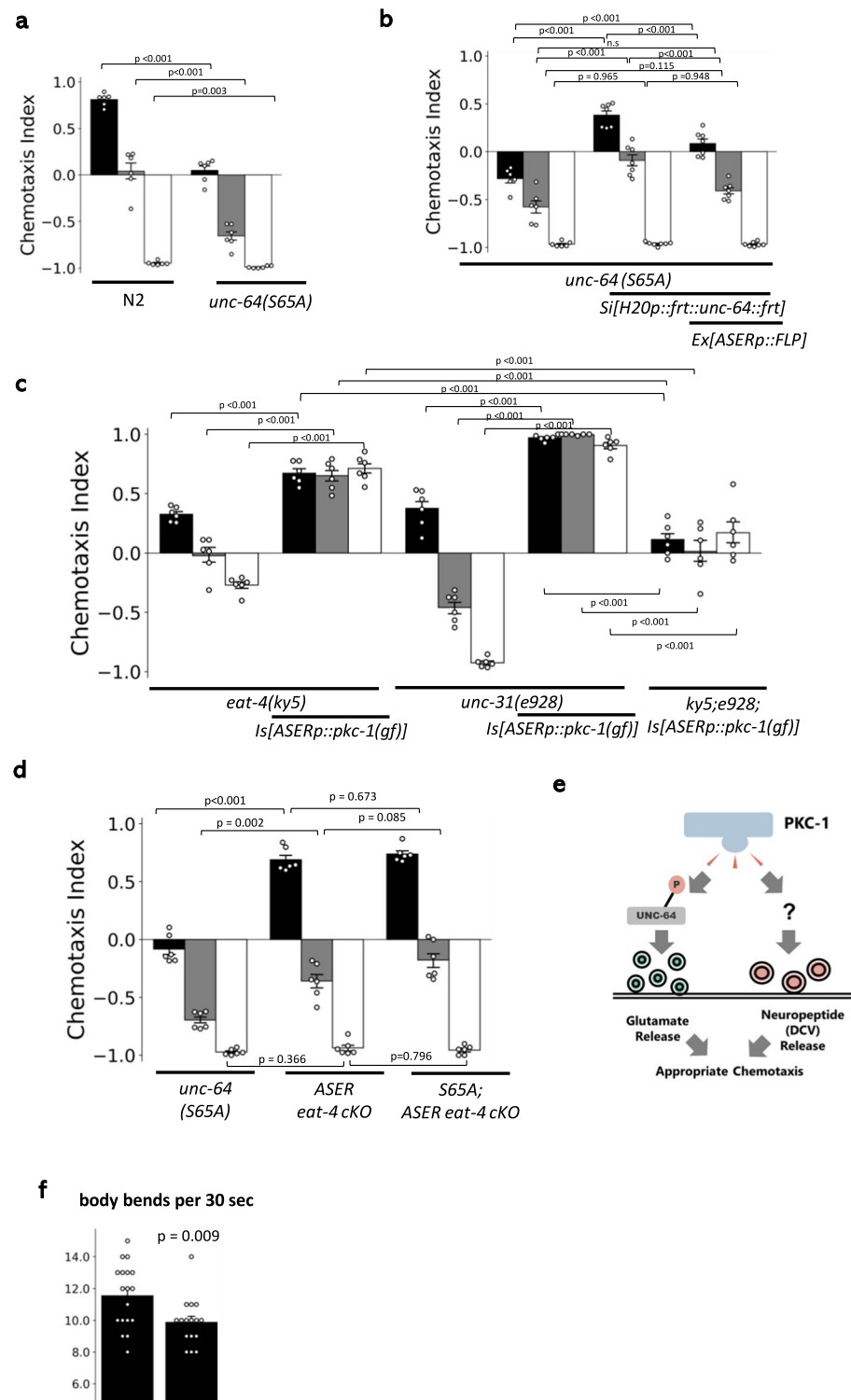

change of NaCl from 50 to 25 mM, which causes ASER activation (Fig. 5a)[30]. First, we observed the response of AIB in *pkc-1(nj3)* and *unc-64(S65A)* mutants using inverse-type calcium probe InversePericam. In *pkc-1(nj3)* and *unc-64(S65A)* mutants, AIB exhibited inhibitory responses to NaCl concentration change (50–25 mM) (Fig. 5b–d). Conversely, when PKC-1(gf) was expressed in ASER,

AIB exhibited excitatory responses when observed with calcium probe GCaMP6s (Fig. 5f). These results suggest that the decreased glutamate release from ASER in *pkc-1(nj3)* or *unc-64(S65A)* mutants is decoded as the reversed response of AIB to NaCl. Moreover, considering that AIB controls reorientation behavior, the direction of AIB response to NaCl decrease, either excitatory or inhibitory,

**Fig. 3 UNC-64 (Ser65) regulates glutamate release to direct navigation behaviors. a** *unc-64* phospho-deficient mutants (*unc-64 Ser65Ala*) tend to migrate toward lower NaCl concentrations. $n = 6$ independent experimental repeats(assays). **b** UNC-64 Ser65 functions in ASER. Migration bias in the *unc-64(S65A)* mutant was rescued by *Si[H20p::frt::unc-64(wt)::frt]*, and the effect of this transgene is suppressed by FLP expression in ASER. $n = 7$ independent experimental repeats(assays). **c** The deficiency of either neuropeptide release (*unc-31(e928)*/CAPS deletion) or glutamate release (*eat-4(ky5)*/vGluT deletion) fails to suppress migration toward higher NaCl induced by ASERp::*pkc-1(gf)*. The double mutants of these genes suppressed *ASERp::pkc-1(gf)*. $n = 6$ independent experimental repeats(assays). **d** Conditional knockout of glutamatergic transmission in ASER suppresses chemotactic bias caused by *unc-64(S65A)*. Genomic *frt::eat-4::frt* is excised by *ASERp::FLP*[27]. $n = 6$ independent experiments(assays). **e** An illustration of PKC-1 signaling. PKC-1 regulates both neuropeptidergic transmission and glutamatergic transmission to drive navigation behavior, and glutamatergic transmission is regulated via the phosphorylation of UNC-64 Ser65. **f** The locomotion rate (=body bends) of *unc-64(S65A)* mutants. As predicted from panel **a**, *unc-64(S65A)* mutants show a defect in locomotion. $n = 18$ (N2), 17 (*unc-64(S65A)*) animals. Error bars indicate SEM for all panels in Fig. 3. *p* values were calculated by the two-sided Welch's test (single comparison) or Tukey's test (multiple comparisons). Source data are provided as a Source Data file.

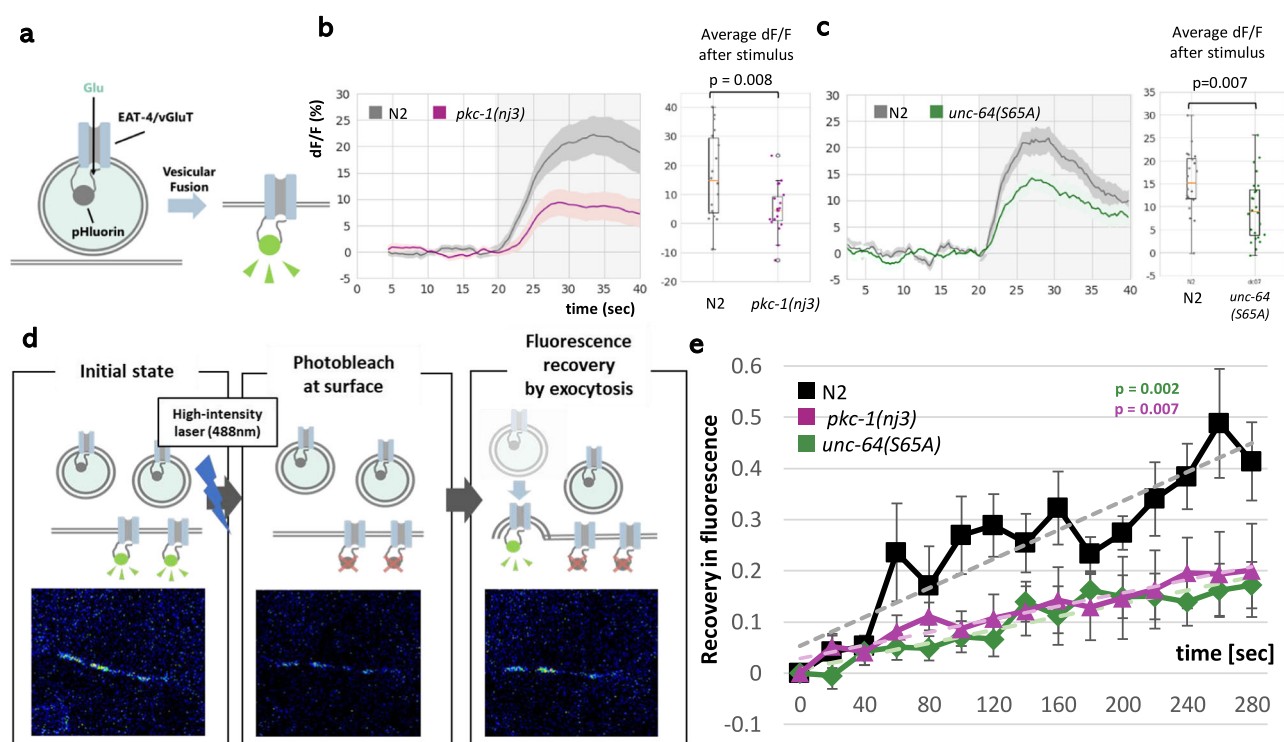

**Fig. 4 Glutamate release from ASER is reduced in *pkc-1(nj3)* and *unc-64(S65A)*. a** A schematic illustration of the fluorescent reporter of glutamate release, *eat-4*::pHluorin, used in B and C[22]. Release of glutamatergic synaptic vesicles causes increased fluorescence. **b** Glutamate release is reduced in *pkc-1(nj3)* mutants. $n = 20$ independent animals. **c** Glutamate release is reduced in *unc-64(S65A)* mutants. $n = 20$ independent animals. *p* values were calculated by the two sided Welch's test (single comparison) or Tukey's test (multiple comparisons). The gray vertical shade in **b**, **c** shows the decrease in NaCl concentration applied to the worm nose (50–25 mM). In **b**, **c**, left, the lines with the shade show the average and SEM, respectively, of changes in pHluorin fluorescence, and in **b**, **c**, right, the boxes extends from 25th to 75th percentile, median is marked by the line, and whiskers indicate the minimum and the maximum values in 1.5× the inter-quartile range. **d** A schematic illustration of the FRAP of ASERp::*eat-4*::pHluorin. EAT-4::pHluorin was photobleached using high-intensity 488 nm Argon laser until the fluorescence decreased to ~50% of the original level. Once fluorescent EAT-4::pHluorin at the surface membrane is photobleached, the fluorescence is recovered according to the release of synaptic vesicle containing EAT-4::pHluorin. Since worms do not experience any change in NaCl concentration during image acquisition, this recovery in fluorescence reflects the baseline release of glutamate. **e** The result of FRAP in wild-type animals, *pkc-1(nj3)* and *unc-64(S65A)* mutants (*n* = 8 independent animals). Error bars indicate SEM. The time course of fractional recovery was linearly fitted. *p* value were calculated by t values of interaction between time course and strains in generalized linear model with Bonferroni's correction. Source data are provided as a Source Data file.

parallel the direction of behavioral bias during navigation observed in these mutants.

**Different sensitivities between excitatory and inhibitory glutamate receptors decode the context encoded as the basal level of glutamate release.** However, how does a mere reduction in presynaptic transmission in *pkc-1(nj3)* and *unc-64(S65A)* mutants result in reversal rather than attenuation of the AIB response? Reduced level of glutamatergic transmission in UNC-64(S65A) appears to actively drive migration to low NaCl concentrations

(Fig. 3d). In *C. elegans*, presynaptic transmission occurs in the following process: unstimulated sensory neurons have a certain amount of neurotransmitter released from presynapse (basal release), which gradually increases or decreases based on sensory perception (stimulus-induced change in release)[22,33,34]. The difference in the magnitude of stimulus-induced glutamate increase in the mutants is unlikely to cause the reversal of AIB responses, because the direction of migration behavior is independent of the magnitude of change in the perceived NaCl concentration[4].

Therefore, we focused on basal glutamate release from ASER, which is irrelevant to changes in perceived NaCl on gradient. In

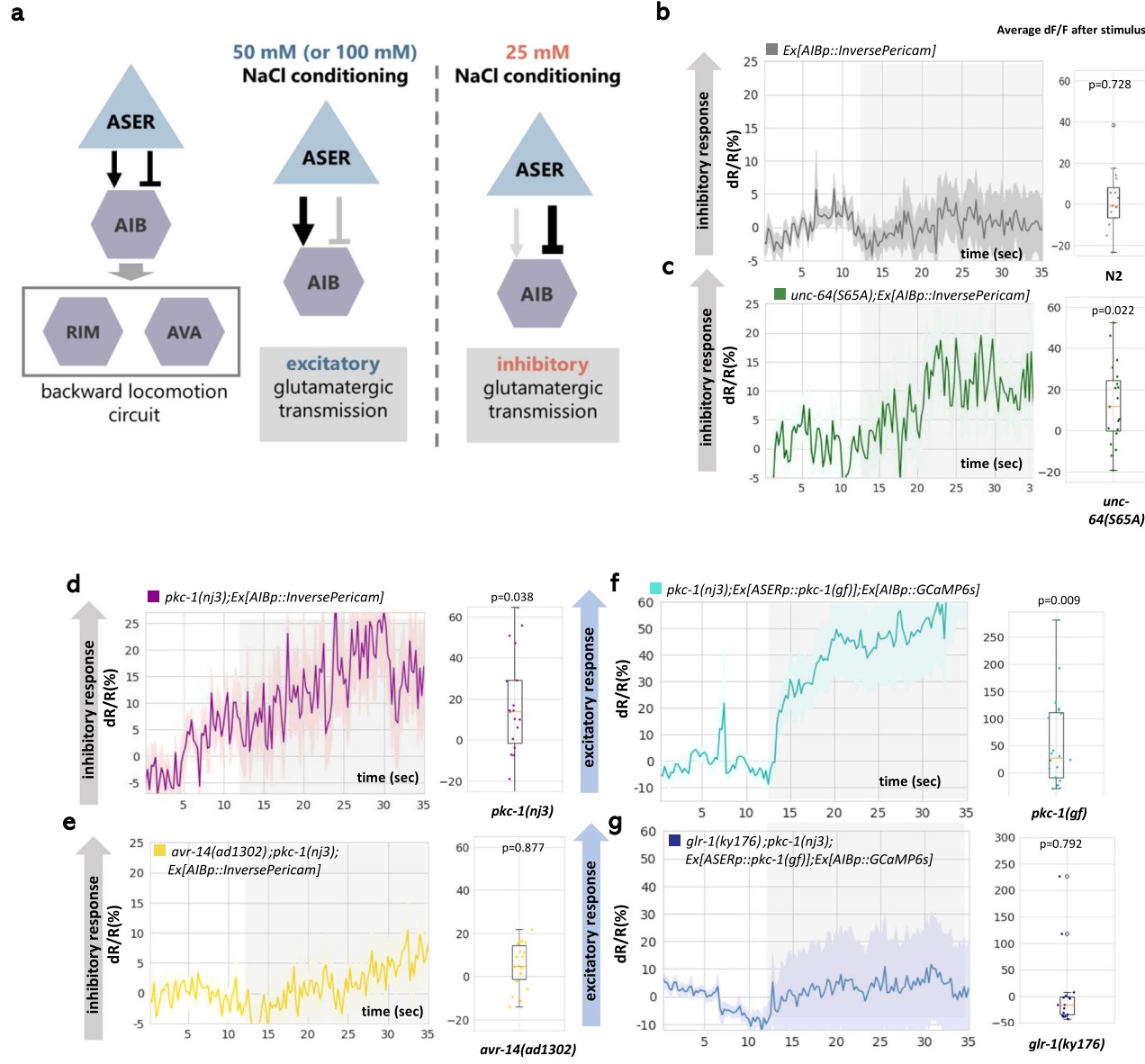

**Fig. 5 Response of AIB interneuron to NaCl concentration change is reversed by PKC-1 and UNC-64 Ser65. a** Summary of the study by Sato et al., 2021. AIB responds to a change in NaCl concentration via glutamatergic input from ASER. Furthermore, the synaptic transmission between ASER and AIB can be excitatory or inhibitory, according to the NaCl concentration experienced in the conditioning phase. **b–d** The response of AIB to NaCl concentration change from 50 to 25 mM measured in freely moving animals by the same setting as Sato et al., 2021 (see "Methods"). Inverse-Pericam, an inverse-type fluorescent calcium probe was used for imaging. An inhibitory response was observed in *pkc-1(nj3lf)* and *unc-64(S65A)* mutants. **e** The inhibitory response caused by *pkc-1(nj3lf)* was diminished in the mutant of inhibitory glutamate receptor, *avr-14 (ad1302)*. **f** AIB exhibited an excitatory response in *pkc-1(nj3);Ex[ASERp::pkc-1(gf)]*. The calcium probe GCaMP6s was used for the imaging. **g** The excitatory response caused by ASERp::*pkc-1(gf)* was diminished in the mutant of the excitatory glutamate receptor, *glr-1(ky176)*. In **b–g**, the vertical gray shading indicates a decrease in NaCl concentration (50–25 mM) in solution surrounding the animals in the microfluidic device. The lines with the shade(error band) show the average and SEM, respectively, of changes in GCaMP or Inverse-Pericam fluorescence. In the right panels, the boxes extends from 25th to 75th percentile, median is marked by the line, and whiskers indicate the minimum and the maximum values in 1.5× the inter-quartile range. p values were calculated by one-sample Welch's test(two-sided). n = 17 (N2), 19 (*unc-64(S65A)*), 19 (*pkc-1(nj3)*), 22 (*avr-14(ad1302); pkc-1(nj3)*), 20 (ASERp::*pkc-1(gf)*), and 16 (ASERp::*pkc-1(gf)*; *glr-1(ky176)*), independent animals, respectively. Source data are provided as a Source Data file.

*pkc-1* and *unc-64* mutants, the glutamate level at the synapses is likely reduced at all times. For confirmation, we observed the basal release level using fluorescence recovery after photobleaching (FRAP) of vGluT-pHluorin[35]. In this FRAP experiment, only vGluT-pHluorins on the outside surface of axonal membrane are bleached, and the fluorescence recovery is caused by translocations of vGluT-pHluorin from inside the axon to the surface (Fig. 4d). Thus, FRAP reflects the rate of translocation to the

surface, i.e., exocytosis events. We photobleached ASERp::*eat-4*::pHluorin to ~50%, and measured a rate of fluorescence recovery for ~5 min in the absence of any change in NaCl concentration[35]. As expected, the rate of recovery is clearly reduced in both *unc-64(S65A)* and *pkc-1(nj3)* mutants compared to wild-type animals (Fig. 4e). ASERp::*pkc-1(gf)*, in turn, does not significantly increased the basal release (Fig. S6), suggesting that wild-type worms have sufficient amount of baseline release to

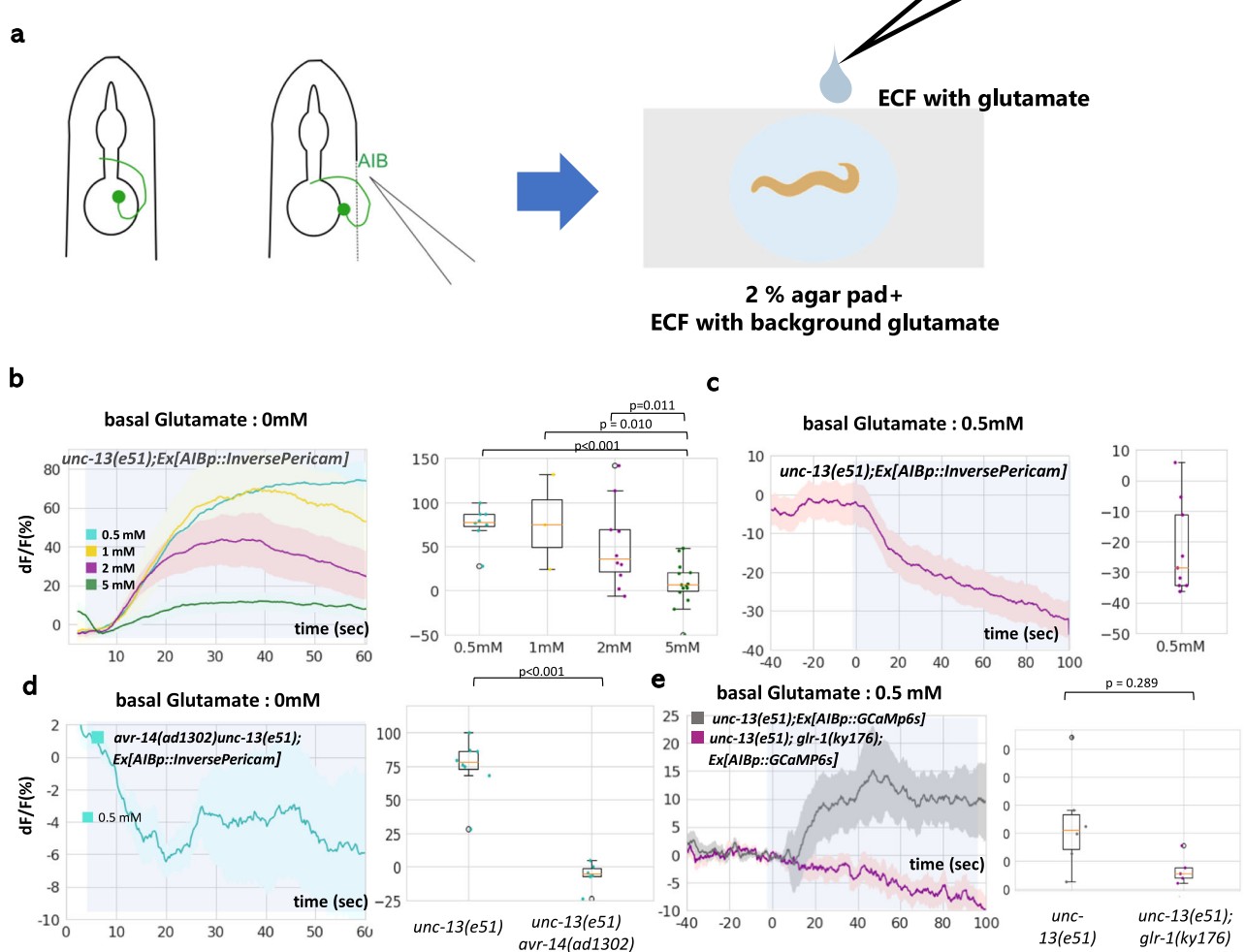

**Fig. 6 Basal glutamate release is decoded as the reversed postsynaptic response. a** Direct application of glutamate to the AIB neurons. *unc-13(e51)* mutants, which do not have synaptic inputs to AIB, were used to exclude the indirect effects of glutamate application. Inverse-Pericam and GCaMP6s were used in the experiments. **b** Inhibitory responses were evoked by glutamate application without preexposure to glutamate (0 mM background). 0.5, 1, 2, and 5 mM of glutamate was applied to $n = 8$, 3, 10, and 10 independent animals, respectively. **c** 2 mM glutamate was applied after preexposure to background glutamate (0.5 mM). Reversed response of AIB (evoked excitatory response) was observed. $n = 8$ independent animals. **d** The inhibitory response (to 0.5 mM glutamate) was abolished entirely in *avr-14* mutants. $n = 6$ independent animals. **e** The excitatory response (to 2 mM glutamate with 0.5 preexposure) was abolished entirely in *glr-1* mutants. $n = 6$ independent animals. In **b–e**, right, the boxes extends from 25th to 75th percentile, median is marked by the line, and whiskers indicate the minimum and the maximum values in 1.5× the inter-quartile range. *p* values were calculated by the Welch's test (single comparison) or Tukey's test (multiple comparisons). Source data are provided as a Source Data file.

induce excitatory response of AIB, consistent with previous observation of AIB responses[30].

From these results, we designed a model described in Fig. 7d. We hypothesized that excitatory and inhibitory glutamate receptors have different sensitivities. If so, the difference in basal glutamate levels can be interpreted as an excitatory or inhibitory response (Fig. 7d).

We tested this possibility by direct exposure of AIB to glutamate (Fig. 6a). We glued the animal onto agar pad by cyanoacrylate glue and covered the animal by extracellular fluid with background glutamate. Then, AIB neurons were dissected out from the body using a glass microneedle. After imaging of AIB expressing InversePericam or GCaMP6s were started, AIB was exposed to additional glutamate. To exclude indirect effects of glutamate via other neurons, we used neurotransmission-defective mutant *unc-13(e51)*[36].

In this experiment, AIB exhibited inhibitory responses to application of 0.5–5 mM glutamate at the concentrations tested (Fig. 6b), where the amplitudes of responses *decreased* when glutamate concentrations increased. This suggests that receptors

that mediate inhibitory and excitatory AIB responses have different sensitivity to glutamate. Next, we pre-exposed the neurons to 0.5 mM glutamate to mimic a high basal glutamate level, and then exposed them to additional 2 mM glutamate. Consistent with the model, AIB exhibited an excitatory response, which was inverse to the effect of 2 mM glutamate without preexposure (Fig. 6c).

Next, we examined receptors that mediate inhibitory and excitatory responses. AIB is known to express excitatory and inhibitory glutamate receptors. Context-dependent response of AIB to NaCl concentration change is known to be mediated by GLR-1, glutamate-gated cation channel, and AVR-14, glutamate-gated chloride channel (GluCl), respectively[30]. AVR-14 likely functions as homomers, because a deletion of the other GluCl gene expressed in AIB, *glc-4*, does not impair AIB response to NaCl[30]. Indeed, inhibitory calcium responses of AIB in *pkc-1(nj3)* and *unc-64(S65A)* mutants were abolished by the *avr-14(ad1302)* mutation (Fig. 5e). On the other hand, excitatory response caused by ASERp::*pkc-1(gf)* was abolished by *glr-1(ky176)* mutation

(Fig. 5g). GLR-2 can potentially form heteromeric receptor with GLR-1 in AVA interneuron[37], and glr-2 is expressed also in AIB neurons. However, in contrast to glr-1, glr-2 mutant exhibited obvious excitatory response to NaCl (Fig. S7), suggesting that GLR-1 might function as homomeric receptor in AIB neurons. Consistent with these data, inhibitory response of AIB to 0.5 mM glutamate was abolished in avr-14(ad1302) mutant (Fig. 6d), while excitatory response to 2 mM glutamate with 0.5 mM preexposure was eliminated in glr-1(ky176) mutants (Fig. 6e).

Therefore, the direction of AIB response is determined by glutamate levels prior to its increase, and this response was mediated by excitatory receptor GLR-1 and inhibitory receptor AVR-14.

**Excitatory glutamate receptor GLR-1 and inhibitory glutamate receptor AVR-14 have different sensitivities.** Our model hypothesized that inhibitory and excitatory receptors have different sensitivities. To test this hypothesis, we heterologously expressed GLR-1 and AVR-14 in Xenopus oocytes. Oocytes were clamped to −50 to −60 mV, and the current response to glutamate at each concentration was recorded. Consistent with previous studies, both receptors show inward current to ample concentration of glutamate[37–40]. Note that AVR-14 shows inward chloride current due to high Cl− concentration in oocytes[39]. AVR-14 shows a nearly maximum response to 0.2–0.5 mM glutamate application, while GLR-1 did not show an obvious response to 2 mM or lower glutamate (Fig. 7a–c). Therefore, the sensitive concentration range of AVR-14 is lower than that of GLR-1.

We also analyzed desensitization time constant of these receptors (Fig. S8a). In this analysis, though AVR-14 might exhibit slower desensitization dynamics relative to GLR-1, the difference was not statistically significant, and both receptors are desensitized at least within a minute. This suggests that under high basal glutamate concentration (0.5–2 mM in Fig. 7b, c), the sensitive AVR-14 is already activated and thus desensitized within seconds, while GLR-1 is not activated and thus cannot desensitize at this concentration. In addition, we roughly observed the localization of the two receptors in AIB neurons. The two receptors did not appear to localize at a specific region of the axon, and the pattern are not obviously different from each other (Fig. S8b).

Taken together, basal glutamate level could be decoded as reversed responses of AIB neurons to glutamate increase through different sensitivities of GLR-1 and AVR-14.

## Discussion
The plastic navigation depending on context is an indispensable ability for animals to reach a favorable location, but how this plasticity arises from molecules and interpreted in neurons to generate appropriate behavior has not been clearly explained. Our genetic, proteomic, physiological, and behavioral analyses provide an architecture of bidirectional navigation behavior in C. elegans (Fig. 8). The switch in migration direction is set out by presynaptic phosphorylation of UNC-64/Syntaxin 1A to cause a change in basal glutamate release level, and decoded as a bidirectional postsynaptic response via the differences in sensitivity of excitatory and inhibitory receptors.

In this study, it was suggested that there is a difference in the sensitivity of the two receptors: AVR-14 is more sensitive than GLR-1. Previous studies have shown that glutamate-gated cation channels (AMPA receptors) and GluCls have completely different glutamate binding sites and molecular kinetics[41–43]. However, it is likely that the sensitivity is unique to each molecule rather than to each family: for example, for a

molecule related to AVR-14, AVR-15, a dose-response to glutamate in Xenopus oocyte was measured in a similar experimental setting to the present study. Its EC50 is around 2 mM and thus is much less sensitive compared to AVR-14, indicating that a receptor sensitivity is not necessarily determined for the receptor family.

In this study, we provided a neuron-targeted phosphoproteomics dataset in C. elegans. Figures 2 and S2 show that the classical phosphoproteomic analysis using whole-body extraction does not necessarily capture neural events. This suggests re-analysis of previous proteomic studies using the technique reported in this study would improve the understanding of each biological process in the viewpoint of neuroscience. Furthermore, a number of research in C. elegans shows intriguing functions of kinases in the nervous system[44–46], nevertheless, a limited number of downstream targets were identified. Our neuron-targeted phosphoproteomic protocol would significantly contribute to improving investigations downstream of kinases, thus empowering the neuroscientific research through C. elegans. Indeed, we found phosphorylation of UNC-64/Syntaxin 1A Ser65 specifically regulates release of synaptic vesicles. Since amino acid sequences around this site are highly conserved among Syntaxin homologs in various species and are phosphorylated (by unknown kinases) at least in mammals (Fig. S3), function of the phosphorylation might also be conserved. This result suggests the findings by our approach in C. elegans leads to fundamental understandings of signaling pathway and its function in the nervous system of animals.

The type of synaptic plasticity revealed in this study might form the fundamental basis of navigation mechanisms in C. elegans. Though AIB is not the only interneuron that receives synaptic input from ASER[47], AIB shares similar features with the other postsynaptic interneurons. Indeed, another major postsynaptic interneuron, AIY, also expresses inhibitory and excitatory glutamate receptors[48]. Moreover, a recent study on thermotaxis demonstrated the bidirectional response of AIY to thermal changes according to the current and experienced temperature[49].

The key principle of the mechanism we found in this report is that changes in presynaptic baseline release generate biphasic postsynaptic responses. This is substantially different from classical mechanisms of synaptic plasticity which typically leads to synaptic potentiation or attenuation[50]. However, this can be conserved in other species, because (1) the phosphorylation of Ser65 at Syntaxin 1A, either by PKC-1 or other kinases, is conserved in mammals (Fig. S3), (2) some synapses also show graded transmission in mammals[51], (3) There are numerous examples of receptors that share the same ligands despite having different sensitivities. Overall, our results both provide comprehensive understanding of bidirectional navigation and pioneer frontiers in the plastic regulation of synaptic transmission.

## Methods
**C. elegans strains and culture**. Animals were grown at 20 °C on NGM plates seeded with E.coli[52]. Bristol N2 strain was used as the wild-type C. elegans. The animals were fed with E. coli strain NA22 in behavioral experiments to avoid starvation and fed OP50 strains in imaging experiments to suppress intestinal autofluorescence.

Strains used in this study are:

Bristol N2, pkc-1(nj3), pkc-1(nj3);Ex[gcy-5p::pkc-1], pkc-1(nj3);Ex[gcy-5p::pkc-1(A160Egf)], Is[H20p::pkc-1(gf);unc-122p::mcherry], Is[rgef-1p::TurboID;myo-3p::venus];Is[H20p::pkc-1(gf);unc-122p::mcherry], nj3;Is[rgef-1p::TurboID;myo-3p::venus], unc-64([Ser65Ala]), Si[H20p::frt::unc-64(WT)::frt::let-858 3′ UTR::GFP::unc-54 3′UTR;Cb unc-119];unc-64(pe[Ser65Ala]), Ex[gcy-5p::nFLP;unc-122p::mcherry], Ex[gcy-5p::eat-4::pHluorin;lin-44p::mcherry], pkc-1(nj3);Is[gcy-5p::eat-4::pHluorin;lin-44p::mcherry], Is[gcy-5p::eat-4::pHluorin;lin-44p::mcherry], unc-64(S65A);Is[gcy-5p::eat-4::pHluorin;lin-44p::mcherry], eat-4(ky5);Is[gcy-5p::pkc-1(gf)], unc-31(e928);Is[gcy-5p::pkc-1(gf)], ky5;e928;Is[gcy-5p::pkc-1(gf)], kySi76kySi77[frt::eat-4genomic CDS::frt::let-858 UTR::mcherry::eat-4 genomic 3′ UTR];Is[gcy-5p::nFLP;unc-122p::venus], unc-64(S65A);kySi76kySi77[frt::eat-4

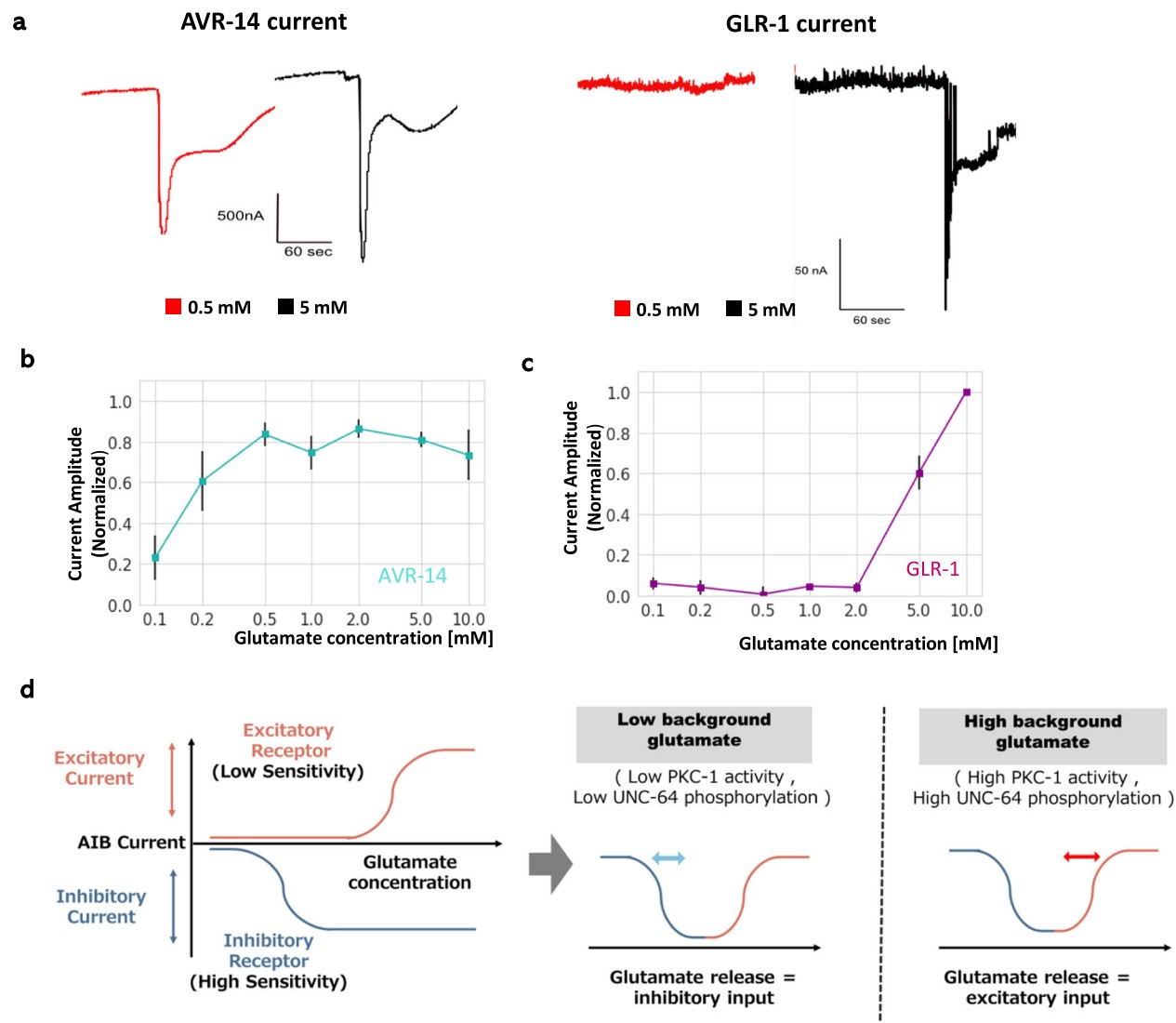

**Fig. 7 The differential sensitivity between the excitatory and inhibitory glutamate receptors, which leads to reversal of postsynaptic response.**
**a** Examples of the current response to 0.5 and 5 mM glutamate. When the oocyte was clamped to −50 to −60 mV, both chloride and cation (mainly sodium) currents were observed as inward currents[37–40]. While the response to 0.5 mM glutamate is comparable to that to 5 mM glutamate in AVR-14-expressing oocytes, almost no current was observed in GLR-1(+STG-1, SOL-1)-expressing oocytes in response to 0.5 mM glutamate. **b** Dose-response relationship between glutamate concentration and normalized current in AVR-14-expressing *Xenopus* oocyte. n = 5 independent oocytes. **c** Dose-response relationship of GLR-1-expressing *Xenopus* oocyte. Functional GLR-1 glutamate receptor was reconstituted by expressing *glr-1* along with auxiliary proteins, *sol-1* and *stg-1*[38]. n = 5 independent oocytes. **d** Schematic illustration of the bidirectional response of AIB to glutamate via excitatory and inhibitory receptors with different sensitivities. When these receptors have different sensitivity ranges (left), the total current is expected to become well-shaped, where depending on basal level of glutamate the postsynaptic response to additional glutamate becomes either inhibitory (middle) or excitatory (right). The vertical lines in **b**, **c** show SEM. Source data are provided as a Source Data file.

genomic CDS::frt::let-858 UTR::mcherry::eat-4 genomic 3′UTR];peIs[gcy-5p::nFLP;unc-122p::venus], Ex[npr-9p::InversePericam216a;npr-9p::mcherry;lin-44p::venus], unc-64(S65A);Ex[npr-9p::InversePericam216a;npr-9p::mcherry;lin-44p::venus], pkc-1(nj3);Ex[npr-9p::InversePericam216a;npr-9p::mcherry;lin-44p::venus], avr-14(ad1302);pkc-1(nj3);Ex[npr-9p::InversePericam216a;npr-9p::mcherry;lin-44p::venus], pkc-1(nj3);Ex[gcy-5p::pkc-1(gf)];Ex[npr-9p::GCaMP6s;npr-9p::mcherry;lin-44p::venus], glr-1(ky176);pkc-1(nj3);Ex[gcy-5p::pkc-1(gf)];Ex[npr-9p::GCaMP6s;npr-9p::mcherry;lin-44p::venus], unc-13(e51);Ex[npr-9p::InversePericam216a;lin-44p::venus], unc-13(e51);Ex[npr-9p::GCaMP6s;npr-9p::mcherry;lin-44p::venus];Ex[npr-9p::InversePericam216a;lin-44p::venus], unc-13(e51);glr-1(ky176);Ex[npr-9p::GCaMP6s;npr-9p::mcherry;lin-44p::venus], unc-64([Ser61Glu]), unc-64([Ser61GluSer65Glu]), unc-64([Ser61A]), Ex[gcy-5p::GCaMP2.0;lin-44p::GFP], pkc-1(nj3);Ex[gcy-5p::GCaMP2.0;lin-44p::GFP], Ex[gcy-5p::pkc-1(gf)];Ex[gcy-5p::GCaMP2.0;lin-44p::GFP], Is[gcy-5p::eat-4::pHluorin;lin-44p::mcherry];Is[gcy-5p::pkc-1(gf);lin-44p::GFP], Is[npr-9p::glr-1::GFP,lin-44], Ex[npr-9p::avr-14::mcherry]

**Generation of transgenic lines**. To generate multicopy transgenic lines, we followed the standard germline transformation protocol[53]. We injected 5–60 ng/μL plasmids along with a co-injection marker and carrier DNA (pPD49.26) into each animal.

We used the CRISPR/Cas9 system to generate variants of the *unc-64* Ser65 site. We injected in vitro reconstructed Cas9-RNA complexes and repair templates (ssDNA) into the animals' gonads[19]. The guide RNA was prepared using Alt-R from Integrated DNA Technologies. The sequence of guide RNA was as follows: 5′-AGAGGUCAAGAAGAAGCAUU-3′ The repair template was prepared using FASMAC. The sequence of the template was as follows: 5′-TTTTAAAACATACT CTGATCGTTAACTGGATTAGCTAAAATTGCAGAATGCTTCTTCTTGACCT CTTCAACATTATTCGC-3′

The edited genome sequence was verified using Sanger sequencing.

To generate the single-copy rescued line of *unc-64*, we used MosSCI to insert the rescue transgenic array into the ttTi5605 locus of EG6699[20]. The excision of *unc-64* ORF by *gcy-5p::nFLP* was confirmed by GFP fluorescence in ASER (10/10 worms).

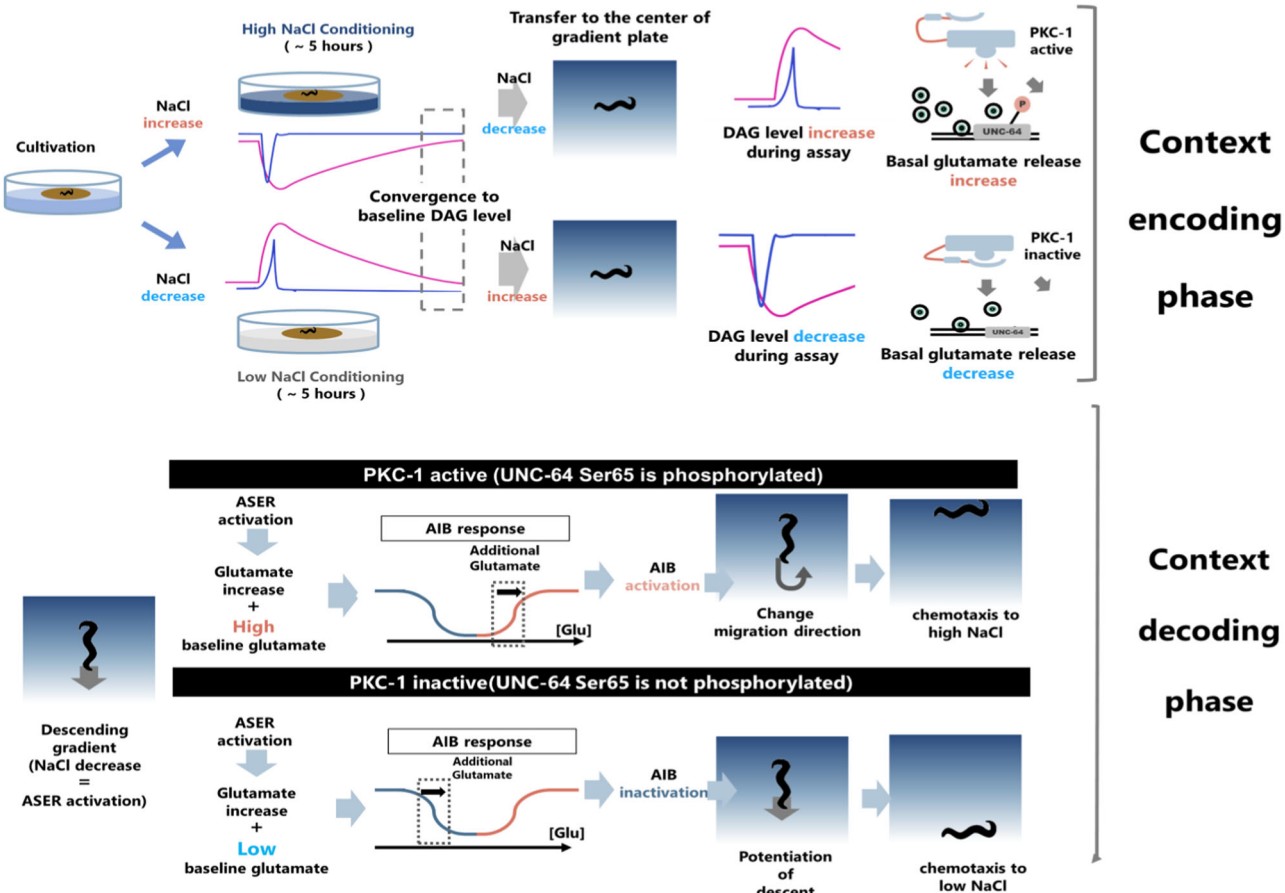

**Fig. 8 Summary of the study.** In the encoding phase (top), when the animals are transferred from the conditioning plates to the center of NaCl gradient in the test plate, ASER is activated or deactivated(blue line), and the amount of DAG changes (pink line, as described in Fig. S1). DAG regulates the activity of PKC-1 to alter its behavior. The activity of PKC-1 regulates the phosphorylation level of UNC-64 Ser65, thereby altering the level of baseline glutamate release from ASER. When animals move on the NaCl gradient in the decoding phase (bottom), ASER responds to the change in NaCl concentration. When worms perceive a NaCl decrease, ASER is activated, and glutamate release from ASER increases acutely. Baseline release is determined by PKC-1 activity in the encoding phase and occurred independently of this increase. When the baseline release rate is low, the highly sensitive inhibitory glutamate receptor in the AIB interneuron responds to glutamate increase (see Fig. 7). When the baseline release rate is high, the low-sensitivity excitatory glutamate receptor in the AIB interneuron responds to an increase in glutamate, where the inhibitory receptor is saturated (or already desensitized) and cannot respond to glutamate increase (see Fig. 7). Therefore, AIB interneuron is activated by a decrease in NaCl concentration when the baseline release rate is high and deactivated when the baseline is low. Since the activation of AIB causes a change in the direction of migration through reorientation behavior, animals with high basal glutamate release rates migrate toward high NaCl concentrations, whereas animals with low basal glutamate release rates migrate toward low NaCl concentrations.

To generate AVR-14-fused mCherry, we used a fusion PCR protocol: templates were *npr-9p::avr-14*[30] and mCherry was cloned into pPD49.26 backbone. Primer sequences were as follows: *npr-9p::avr-14*: Left: GCTACAGTTGGGTTGATGAC GACG, Right: TTCACCCTTTGAGACCATATCACGGCTCTGTTTCACATATG CAAC. mcherry: Left: ACAGAGCCGTGATATGGTCTCAAAGGGGTGAAGAA GATAACATGG, Right: ACGGCCGACTAGTAGGAAACAGTTATG

**Plasmid construction**. We used Gateway Technology (Invitrogen) to generate plasmids containing promoters and genes. We cloned genes of interest into pDEST vectors using the In-Fusion HD cloning kit (Clontech). We then inserted the promoter sequence through the LR reaction, using the following sequence as the cell/tissue-specific promoters. ASER: *gcy-5*, AIB: *npr-9*, nervous system: H20/*limb-1* or *rgef-1*.

**Behavioral assay**. Chemotaxis assay was performed as previously described[4] with slight modifications. Two agar plugs (cylinder, 14.5 mm in diameter, with 150 mM and 0 mM NaCl) were placed onto assay plates (2% agar, 1 mM $MgSO_4$, 1 mM $CaCl_2$, 25 mM potassium phosphate), which were left for 23–25 h to form the NaCl gradient. The animals were grown on standard NGM plates with 50 mM NaCl for 4 days, then transferred to conditioning plates (NGM plates with 100, 50, or 25 mM NaCl) using wash buffer (50 mM NaCl, 1 mM $MgSO_4$, 1 mM $CaCl_2$, 25 mM potassium phosphate, 0.5 g/L gelatin). After the conditioning phase (typically 5–6 h), worms were collected from the plates with wash buffer and washed 2

times. After washing, the animals were placed in the center of the assay plates. Immediately before placing worms on the assay plates, the plugs were removed and 1 μL each of 0.5 M $NaN_3$ were spotted at the centers of the peaks. Worms were allowed to move on the plate for approximately 60 min and then transferred to 4 °C to stop their movement.

The worms in each region on the plate (Fig. 1a) were counted to calculate the following:

Chemotaxis Index = $(N_{high} - N_{low})/(N_{all} - N_{start})$

Locomotion of animals were quantified following previous reports[21] with slight modifications. Animals were transferred from growth plate to fresh NGM plates without *E. coli*, and were acclimated for <5 min. Then, animals were transferred to another NGM plate to completely remove bacteria and were acclimated for <1 min. Body bends in 30 sec were measured by manual counting under a dissecting microscope.

**Phosphoproteomic analysis**. For phosphoproteomic analysis from whole-body extraction, we used 10 plates (~1000 age-synchronized young adult animals per plate) per replicate. We used 40–60 plates per replicate for neuro-phosphoproteomic analysis using TurboID.

First, we lysed the animals in PTS buffer (100 mM $NH_4HCO_3$, 12 mM sodium deoxycholate, 12 mM sodium N-lauroylsarcosinate) by sonication. Biotin-labeled samples were enriched by High Capacity Streptavidin Agarose (Pierce). Samples were incubated with the beads in PTS buffer for 1 h and then washed four times with PTS buffer. Each sample in the PTS buffer was reduced with 10 mM DTT at

60 °C for 30 min and then alkylated by incubation with 22 mM iodoacetamide in darkness at 37 °C for 30 min. The resultant protein sample was diluted 1:5 with 100 mM $NH_4HCO_3$ solution and digested with 0.4 µg trypsin (Roche) by incubation at 37 °C for 18 h in darkness. After digestion, an equal volume of ethyl acetate was added to the sample, and the mixture was acidified with 0.5% TFA and mixed well to transfer the detergents into an organic phase. After the sample was centrifuged at $15,700 \times g$ for 1 min at room temperature, an aqueous phase containing peptides was collected. The sample was concentrated using a centrifugal evaporator (EYELA) and desalted using a MonoSpin C18 column (GL Sciences). The eluate was dried prior to the LC-MS/MS analysis.

The dried and desalted peptides were dissolved in distilled water containing 2% acetonitrile and 0.1% TFA. LC-MS/MS analyses were performed using a mass spectrometer (Q Exactive Plus, Thermo Fisher Scientific) equipped with a nano UHPLC system (Dionex Ultimate 3000, Thermo Fisher Scientific). The peptides were loaded onto the LC-MS/MS system with a trap column (0.3 × 5 mm, L-column, ODS, Chemicals Evaluation and Research Institute) and a capillary column (0.1 × 150 mm, L-column, ODS, Chemicals Evaluation and Research Institute) at a flow rate of 20 µL/min. The loaded peptides were separated by a gradient using mobile phases A (1% formic acid in distilled water) and B (1% formic acid in acetonitrile) at a flow rate of 300 nl/min (0% B for 5 min, 0–30% B for 150 min, 30–50% B for 10 min, 50–95% B for 0.1 min, 95% B for 9.8 min, 95–0% B for 0.1 min, and 0% B for 5 min). The eluted peptides were electrosprayed (2.0 kV) and introduced into the MS equipment (positive ion mode, data-dependent MS/MS). The top 10 most intense precursor ions were isolated and fragmented by higher collision energy dissociation with normalized collision energy (27%). For full MS scans, the scan range was set to 350–1500 m/z at a resolution of 70,000, and the AGC target was set to 3e6 with a maximum injection time of 60 ms. For MS/MS scans, the precursor isolation window was set to 1.6 m/z at a resolution of 17,500, and the AGC target was set to 5e5 with a maximum injection time of 60 ms. The Orbitrap mass analyzer was operated with the "lock mass" option to perform shotgun detection with high accuracy. The raw spectra were extracted using Proteome Discoverer 2.2 (Thermo Fisher Scientific) and searched against the *C. elegans* UniProt database (TaxID = 6239 and subtaxonomies, v2019-08-05) with following settings: The parameter of the cleavage was set to trypsin, and the missed cleavage was allowed up to 2. The mass tolerances were set to 10 ppm for precursor ion and 0.05 Da for fragment ion. As for protein modifications, we set carbamidomethylation (+57.021 Da) at Cys as fixed modifications for peptide, oxidation (+15.995 Da) at Met and phosphorylation (+79.966 Da) at Ser, Thr and Tyr as dynamic (non-fixed) modifications for peptide, and acetylation (+42.011 Da) at amino-terminus as a dynamic modification for protein terminus. The amount of each peptide was semi-quantified using the peak area with Precursor Ions Quantifier in Proteome Discoverer 2.2.

The summary of proteomic analysis are provided in Supplemental Data S1 and S2.

Histological visualization of biotin was performed using streptavidin-conjugated Alexa Fluor 568 (Thermo Fisher). The worms were fixed using Bouin's tube fixation[54]. Images were acquired using a Leica TCS-SP5 confocal microscope.

**Measurement of glutamate release using fluorescence microscopy**. The animals were transferred to NGM plates with 50 mM NaCl in which the osmolarity was adjusted to 350 mM Osm with glycerol on the day before imaging. Animals were set into a microfluidic PDMS chip[55] filled with imaging buffer (50 mM NaCl, 1 mM $MgSO_4$, 1 mM $CaCl_2$, 25 mM potassium phosphate). The osmolarity of each buffer was adjusted to 350 mOsm using glycerol. The stimulus (imaging buffer with 25 mM NaCl) was delivered to the tip of the animal's noses. Images were acquired at 2 or 4 fps with a DMI-6000B microscope (Leica) equipped with an HCX-PLAPO 63× objective (NA, 1.40) and Metamorph(ver.7.7.3).

The fluorescence intensity relative to F0 (dF/F) was calculated for time-lapse images, where the average fluorescence intensity of GCaMP over 10 s prior to stimulus was set as *F*0. The 2.5 s rolling average is shown in Fig. 1b, c. Average d*F*/*F* over 15 s after stimulation was used for quantification.

For the quantification of basal release, we used FRAP of pHluorin[35]. Animals were placed on 5% agarose pad with wash buffer containing 0.5 mM tetramisole and then observed using Leica TCS-SP5 confocal microscope with ×60 objective lens. First, we acquired the image of the axonal region of ASER, and then photobleached *eat-4*::pHluorin using a 488 nm Argon laser at the highest intensity. After photobleaching (typically to ~ 50%), timelapse images were acquired at 0.05 fps (20 s/frame) for 280 s. Afterward, using Fiji(1.5.2p), we quantified the fluorescence of the selected axonal region. We calculated fractional recovery in fluorescence by [ft −fp]/[fi −fp], where fi is the *eat-4*::pHluorin fluorescence before photobleaching, fp is the fluorescence after photobleaching, and ft is the fluorescence at time *t* after photobleaching.

**Imaging of AIB in freely moving animals**. Imaging was performed with an automated tracking system as previously described[30]. Adult animals were washed out from the NGM plates (with 50 mM NaCl) and loaded into the microfluidic arena[30,32]. Changes in the NaCl concentration were delivered by switching the imaging solutions described above. Images were acquired using a BX51 microscope

(Olympus) equipped with a 20× objective and ImagEM EM-CCD camera at 4 fps, U-MF2 filter cube with an FF493/574 dichroic mirror, FF01-498 short-pass filter, Metamorph(ver.7.7.3). Images taken with near-IR illumination were used to keep the animals in the center of the field of view. The ratio of GCaMP/mCherry (*R*) relative to *R*0 was calculated (d*R*/*R*), where the average ratio over 10 s prior to the stimulus delivery was set as *R*0. For quantification, the d*R*/*R* in frames between 10 and 20 s after stimulus delivery were averaged. Since d*R*/*R* was normalized according to *R* prior to stimulus, a 1-sample t-test was used to test whether the responses were significantly different from 0. All images with severe tracking failures in the −40 to +100 frames relative to the stimulus delivery were discarded.

**Direct exposure of AIB neurons to glutamate**. The animals were fixed onto a wet 2% agar pad with cyanoacrylate glue. We dropped 10 µL of extracellular fluid (150 mM NaCl, 5 mM KCl, 5 mM $CaCl_2$, 1 mM $MgCl_2$, 10 mM D-glucose, 15 mM HEPES, pH 7.3) on the animals immediately after fixation. The glass needle was inserted on the ventral side of the animals along the longitudinal axis, and the body wall was dissected carefully so as not to injure the ventral neural ganglion. Immediately after the imaging was started, 10 µL of 2× glutamate at each concentration was added. Images were acquired at 2 fps with a DMI-6000B microscope (Leica) equipped with an HCX-PLAPO 63× objective (NA, 1.40), Metamorph(ver.7.7.3). For imaging with background glutamate, imaging was started at least 4 min after background glutamate application to exclude the acute response to the background. Animals were exposed to additional glutamate 75 s after the start of imaging. Images with severe defocus or obvious injury to the AIB were discarded.

**Electrophysiology**. The voltage clamp recording of Xenopus oocytes was performed as previously described[38,39] with slight modifications. Adult *Xenopus laevis* were acquired from Xenopus Aquaculture Materials. The oocytes were extracted from anesthetized *Xenopus laevis*. cRNA of AVR-14, GLR-1, SOL-1, and STG-1 was synthesized using the mMessage in vitro transcription kit (Ambion). Approximately 50 nL of the cRNA solution (containing 100 ng/µl cRNA) was injected into each oocyte. After 3–4 days of incubation at 17 °C, voltagec clamp measurements were performed using oc-725c (Warner Instruments). The two electrodes were inserted into oocytes, and oocytes were voltage-clamped to −50 to −60 mV in the extracellular solution (100 mM NaCl, 2 mM KCl, 1 mM $CaCl_2$, 2 mM $MgCl_2$, 10 mM HEPES, pH 7.2). Glutamate was applied for 20 s at each concentration (0.1, 0.2, 0.5, 1, 2, 5, and 10 mM, respectively) followed by at least 3 min of wash periods using a perista pump. Dataset was analyzed using Clampfit (10.3.2.1).

**Characterization of UNC-64 function in motor neurons**. Coelomocyte uptake assays were performed as previously described[21]. Young adult animals were immobilized using 10 mM $NaN_3$ on a 5% agar pad and imaged using a Leica TCS-SP5 confocal microscope. The mean fluorescence of 3–4 puncta of the posterior coelomocytes was quantified.

Aldicarb assay was performed as previously described[28]. We placed 25–30 worms on NGM plates containing 1 mM aldicarb. Worms were recorded every 15 min to examine the paralyzed worms. Worms that did not show any behavioral response to three-times touch by a platinum wire picker were counted as paralyzed. Figure S5 shows the average and standard error of the mean of triplicates.

**Image analysis**. We used Metamorph 7.7.3 'track object' tool to track target objects (e.g., cell body) in time-lapse images. Subsequently, fluorescence was quantified using Fiji(1.52p), and analyzed using python(3.7.13).

**Ethics**. Oocyte extraction from *Xenopus laevis* was performed according to a protocol approved by the Animal Ethics Committee of The University of Tokyo (Animal Plan, 20–6).

**Statistics**. All statistical tests were performed using the R scripts. Welch's test was used to compare pairs of strains. For multiple comparisons, one-way ANOVA followed by Tukey's HSD was used. Strains conditioned at the same concentration were compared. All tests were two-sided, whether it is stated or not.

**Reporting summary**. Further information on research design is available in the Nature Research Reporting Summary linked to this article.

## Data availability

The mass spectrometry proteomics data have been deposited to the ProteomeXchange Consortium via the PRIDE partner repository with the dataset identifier PXD031536. The summary of the analysis was provided as Supplemental Data S1 and S2. The *C. elegans* strains generated in this study will be made available upon reasonable request to the corresponding author. Source data are provided with this paper.

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

## Acknowledgements

We thank Masahiro Tomioka for technical advice and providing some of the transgenes; C. Bargmann for providing kySi76kySi77, Motomichi Doi for Ex[*npr-9p::glr-1::GFP*] and Shuzo Sugita for the open syntaxin mutant *unc-64(sks4)*; Some strains were provided by the Caenorhabditis Genetics Center, which is funded by NIH Office of Research Infrastructure Programs (P40 OD010440).

## Author contributions

S.H. designed, interpreted, and performed most of the experiments. S.H. and Y.H. designed and conducted mass spectrometry experiments. H.S. conducted $Ca^{2+}$ imaging of ASER and in vivo imaging of *glr-2* mutants. H.M. conducted *eat-4*::pHluorin experiments using PDMS chips in *pkc-1(nj3)* mutants. Y.H., Y.F., S.K., and C.U. supervised a part of experiments and built instruments. Y.I. supervised the research. S.H. and Y.I. wrote the paper. Each author was funded by: S.H.: 20J13918(JSPS), Y.I.:17H06113 (JSPS), Y.F.: 17H06096(JSPS)

## Competing interests

The authors declare no competing interests.
