## [Peer review file · Nature Communications]

REVIEWER COMMENTS

Reviewer #1 (Remarks to the Author):

Worms migrate up or down the salt gradient, depending on their prior experience. This behavioral plasticity is robust and fascinating, yet the neural and molecular mechanisms are not well understood and somewhat puzzling. In this paper, building on their previous work, the authors have identified UNC-64/Syntaxin as a molecular target of PKC-1, which plays a pivotal role in this behavioral plasticity. They also found that the experience-regulated basal presynaptic glutamate release level at the ASER-AIB synapses together with the distinct glutamate sensitivity of the postsynaptic inhibitory AVR-14 and excitatory GLR-1 receptors can nicely explain the observed salt chemotaxis plasticity.

Overall, this is a very nice piece of work. The authors have collected a large amount of data. The work is also quite comprehensive. It is exciting to see that after years of hard work, the authors have finally come up with a nice model for salt chemotaxis plasticity. I am thus very happy to support its publication in Nature Comms. I only have a few of questions, which I hope the authors will be able to address.

1. The authors based their model on the distinct glutamate sensitivity of AVR-14 and GLR-1 in AIB neuron. This may not be sufficient to explain the observed valence switch at the ASER-AIB synapse. How about the desensitization properties of AVR-14 and GLR-1 toward glutamate? It is possible that while AVR-14 is more sensitive to glutamate, it may also show a more rapid desensitization to glutamate. If true, under higher basal glutamate release, AVR-14 is probably desensitized already, so it cannot respond to further glutamate release from ASER. If so, this mechanism is probably equally, if not more, important. This can be tested.
2. Does UNC-64 show an increased phosphorylation level when conditioned under low salt concentrations vs. high salt concentrations? The model proposed by the authors indicated so.
3. Does pkc-1(gf) promote the basal glutamate release from ASER? The authors only showed that pkc-1(lf) and unc-64(S65A) mutants reduce the basal glutamate release.

Reviewer #2 (Remarks to the Author):

This is an interesting and highly integrative study isolates and defines a new potential molecular determinant underpinning conditioned behaviour in *C.elegans*. The authors have done a good job at both explaining the rationale of various technologies and the potential wider significance of their observations. Given this, it is likely to draw a wider readership from those interested in molecular determinants of behaviour and functional relevance of transmitter release levels. I would highlight the retinal signal processing that speaks to the wider concepts of their work. Indeed, work from Lagnado lab as an example resonates well with the current study.

The overview plays on the idea that DAG or PKC dependent plasticity as mediated by syntaxin phosphorylation is a novel plasticity. I am not sure it is novel but I am sure it has to be of interest.

Given the data leads with the central role of DAG are the authors clear that other DAG targets that underlie presynaptic plasticity are not contributing. Unc-13 for example would be very good at facilitating an augmented transmitter release.

The title is perhaps hyperbole. I imagine the reader would be helped by mentioning salt chemotaxis rather than spatial navigation. Again, the work can afford to be specific without losing interest.

The use of proximity labelling to enrich for neuronal proteins in the biochemically challenging model organism is well conceived. However, the criteria for inclusion based on neuronal (or as sometimes-indicated neural proteins) is not clear. Several proteins found in neurons will be widely expressed and shared by other cell types. The basis of this filtering could be better reported. While appreciating the effort from the first attempts to define phospho proteins using panuronally expressed pkc-1 up and down regulation I am not sure the challenges of this need reporting per se or at least in the main body of the text.

Indeed, taking time to provide a better description of the TurboID approach and the justification for statements like the “dramatic change” on UNC-64 would be useful.

The ideas around syntaxin are interesting and the model that emerges from the work likely to garner wide interest. It would be useful if the authors could comment on whether mutants harbouring the open form of unc-64, which might occlude the PKC-1 dependent effects, have been tested.

Is there are any confounding (motility phenotypes associated with the key syntaxin mutants used in the study. In particular, the extreme aldicarb insensitivity of the unc-64 (S65A) strain raises this issue. One imagines analysis of sub-behaviours that prelude the arrival during salt chemotaxis would be useful in addressing this.

The image-based approaches to synaptic release and functional organization are well used. However, some description or reflection of how well these assays map onto the synaptic parameters the authors suggest are key determinants would be useful.

It is unclear to this referee why the recovery from photo bleaching is a measure of vesicle cycling and not confounded by vesicle mobility. As the latter is known to depend on PKC dependent events. If this is an issue, it seems fair that the authors discuss this limitation to their observations.

The authors are to be given credit for using spritz application of glutamate to exposed postsynaptic neuron. I follow their preferred argument here but equally imagine that receptor properties like activation, desensitization and localization will be important in executing the responses that they see. Particularly given the skilled but limited route of transmitter application. I think a better description of what is known about the receptor elements that are the focus of their work would be useful. I think the shift in response in the *avr-14* and *glr-1* mutants do evidence the contribution of these receptors.

The oocyte expression of single subunits would be more compelling if the authors had evidence that homooligomeric GluCl and AMPA like receptors were consistent with those that are found and used in glutamate transmission at AIB. Published work favours the authors' interpretation about relative affinities at these distinct glutamate-binding sites and building this into their discussion would be fair and wise. Is their previous evidence for robust *glr-1* reconstituted activity?

Minor comments.

The authors have done a good job at creating a guiding narrative but it feels imprecise in places.

The text from Ln 175 to 183 is difficult to interpret. Is the subject of the text the ref 24 or the author's data in Fig 5A.

Ln 184-184 maybe think about the sense of this.

185-191 describes imaged Ca²⁺ and then suggests that reversal response. Is this a cellular or behavioural parameter the authors are describing?

Reviewer #3 (Remarks to the Author):

The study uses an elegant proximity labelling approach to measure the neuronal-specific proteome and phosphoproteome and identify a phosphorylation site in Syntaxin that is functional in *C. elegans* chemotaxis. The promiscuous biotin ligase TurboID was expressed in neuronal cells, causing non-specific biotin labeling of neuronal cell proteins. These proteins were then subjected to phosphoenrichment to identify the neuronal phosphoproteome. Mutants of PKC-1 were compared to identify potential downstream substrates that regulate salt chemotaxis, and the researchers focus on a single candidate, S65 on Syntaxin for further functional analysis. I have been asked to evaluate the proteomics analysis.

1. The authors should deposit mass spectrometry raw into the appropriate repositories and processed/human-legible tabular data into supplementary information. For example, quantitative and identity information for all phosphopeptides and unmodified proteins should be included as supplementary material.

2. For Figs. 2E and 2F, the authors should provide the sequence of the phosphorylated peptides described.

3. Is there a consensus sequence for PKC-1 substrates? Are the phosphorylated peptides showing high pkc-1(gf)/pkc-1(nj3lf) ratios enriched in a consensus sequence motif?

4. It would be useful to highlight the point in the volcano plot (Fig. 2D) representing S65 of syntaxin.

**This is the response letter to
'NCOMMS-21-39787 Molecular encoding and synaptic decoding of the context during
spatial navigation in *C. elegans*'.**

**This response letter consists of two sections.
1. Overview and 2. Point by point response.**

1. Overview.

Firstly, we deeply thank the editor and reviewers for carefully evaluating the manuscript. The comments by the reviewers were helpful in improving our manuscript. We specifically appreciate the editors' flexible response that they provided to us the comments by Reviewers #1 and #2 prior to the assignment of the final reviewer. Following below is a summary of modifications made to our paper. We tried to address the reviewers' concerns as much as possible.

In the revised manuscript, we added results of six newly performed experiments as follows:

1. Desensitization dynamics of AVR-14 and GLR-1 (Fig.S8A)
2. FRAP assay of basal synaptic release in *ASERp::pkc-1(gf)* (Fig.S6).
3. Genetic analysis using open syntaxin mutants (Fig.S4C).
4. Locomotion rate of UNC-64(S65A) (Fig.S5D).
5. Localization patterns of AVR-14 and GLR-1 (Fig.S8B).
6. AIB imaging in mutants of another glutamate receptor subunit(*glr-2*) (Fig.S7).

We newly submitted and deposited the dataset as follows:

1. We deposited raw phosphoproteomic datasets in ProteomeXchange,
Project Accession Number: PXD031536
Project Name: Hiroki_et_al_2022_TurboID_Phosphoproteomics_RawData
Reviewer Log in:
Username: reviewer_pxd031536@ebi.ac.uk
Password: pVtv5GEz
2. We added the summary table of proteomic and phosphoproteomic analysis as Supplemental Table.

**We edited the manuscripts and figures according to the reviewers and Nature
Communication's Editorial Policies .**

1. We modified all the $n < 10$ bar plots, overlaying dot plots representing each raw value of data points.

Accordingly, we found one mistake in sample size of Fig. S4A (n = 4, not n = 6). We sincerely apologize for this.

2. We added a strain list in Methods section.
3. We added explanations and references in response to reviewers' comments.
4. We modified Figure 2D-F according to the reviewer's suggestions.
5. We changed the title from
'Molecular encoding and synaptic decoding of the context during spatial navigation in C. elegans'
to
'Molecular encoding and synaptic decoding of the context during salt chemotaxis in C. elegans'

In addition to the above, we recognized premature and illegible expressions in the manuscript. We carefully modified the manuscript taking care not to change any scientific meanings or conclusions.

2. Point-by-point response to Reviewers' comments.

We thank all the three reviewers, who kindly suggested important points to improve our manuscript.

Our point-by-point responses follow:

Reviewer #1

1.

The authors based their model on the distinct glutamate sensitivity of AVR-14 and GLR-1 in AIB neuron. This may not be sufficient to explain the observed valence switch at the ASER-AIB synapse. How about the desensitization properties of AVR-14 and GLR-1 toward glutamate? It is possible that while AVR-14 is more sensitive to glutamate, it may also show a more rapid desensitization to glutamate. If true, under higher basal glutamate release, AVR-14 is probably desensitized already, so it cannot respond to further glutamate release from ASER. If so, this mechanism is probably equally, if not more, important. This can be tested.

We appreciate the reviewer's discussion, and agree with the reviewer that desensitization needs to be considered. As suggested by the reviewer, we assessed time constants of desensitization of the two receptors in the oocyte expression experiment including additional experiments conducted for revision and added the quantification results in **Fig. S8***. Our interpretation is as follows: under a high basal glutamate level, say 0.5 mM as in the experiment shown in Fig. 7, GLR-1 is not activated and therefore cannot desensitize. It responds to further increases in glutamate, say to 5 mM, because this concentration spans the sensitivity range of GLR-1. On the other hand, AVR-14 is activated by initial application of 0.5 mM glutamate and then desensitizes as shown in Fig. 7A at a time constant quantified in Fig. S8. It does not respond to further increase in glutamate both because the concentration is above the sensitivity range of AVR-14 and also because the receptor has desensitized already (to some extent) at this concentration. We added discussion on desensitization in the manuscript in **Lines 270-275**.

* Here we newly performed oocyte clamp for GLR-1, because the original dataset of GLR-1 seems to be somewhat messy due to longer clamp and relatively low amplitude, thus did not seem to be suitable for the analysis of desensitization). This was described in legend to Fig. S8.

2.

Does UNC-64 show an increased phosphorylation level when conditioned under low salt concentrations vs. high salt concentrations? The model proposed by the authors indicated so.

As the reviewer suggested, It would be both really interesting and important to know the phosphorylation state of UNC-64. However, considering that salt response of UNC-64 phosphorylation is expected to occur only in the salt-sensing ASER neuron, **we found it quite difficult to observe the dynamic phosphorylation at the synapse in the single sensory neuron in vivo.**

In fact, we tried FRET-based imaging using phosphopeptide-binding peptides using two of previously used FRET backbones (ERKy backbone*1, Eevee backbone*2), but unfortunately, neither of them did show fluorescence of the fluorophores(CFP, YFP) and thus change in fluorescence could not be assessed.

*1 Tomida et al., 2012 doi: 10.1126/scisignal.2002983

*2 Komatsu et al., 2011 doi: 10.1091/mbc.E11-01-0072

3.

Does pkc-1(gf) promote the basal glutamate release from ASER? The authors only showed that pkc-1(lf) and unc-64(S65A) mutants reduce the basal glutamate release.

We performed the FRAP experiment in the Is[ASERp::pkc-1(gf)] strain and added the result to the revised manuscript as Supplemental Figure S6. Is[ASERp::pkc-1(gf)] might slightly increase the recovery rate, but the difference was not statistically significant compared to N2. This might be partly due to a dim EAT-4::pHluorin fluorescence in the ASERp::pkc-1(gf) background probably due to simultaneous expression of two transgene in the same neuron. This would have induced a relatively larger variance in ASERp::pkc-1(gf) in **Fig. S6**. Please note that we do not consider this to weaken our model, for WT worms also show excitatory ASER-AIB transmission under the same experimental condition (Sato et al., 2021). We added this rationale in **Lines 225-227**

Reviewer #2:

1

Given the data leads with the central role of DAG are the authors clear that other DAG targets that underlie presynaptic plasticity are not contributing. unc-13 for example would be very good at facilitating an augmented transmitter release.

We thank the reviewer for the remark. In our newly published paper (See Fig. 3 in Hiroki et al., 2022, doi: 10.1073/pnas.2106974119), we applied a potent DAG analog, Phorbol

12-myristate 13-acetate (PMA). PMA induces chemotaxis towards high-salt when applied to N2 as expected, while it does not have this effect on *pkc-1(nj3lf)* at least at the concentration comparable to the chemotaxis assay (0.2 mg/mL). Therefore, in our chemotaxis assay, the major target of DAG is PKC-1. Furthermore, when *tpa-1*, an nPKC isotype closely related to PKC-1, was lost in addition to *pkc-1*, the effect of PMA is completely lost even when extremely high concentration of PMA was applied. Given that kinase domain of TPA-1 can acts as that of PKC-1 (See Fig. 4 of Hiroki et al., 2022) and thus the functional mechanism seems to be common, this clearly indicates the effect of DAG largely depends on nPKCs in salt chemotaxis.

We added citation and description in **Lines 71-72**.

1.5

The title is perhaps hyperbole. I imagine the reader would be helped by mentioning salt chemotaxis rather than spatial navigation. Again, the work can afford to be specific without losing interest.

We modified the title to '**Molecular encoding and synaptic decoding of the context during salt chemotaxis in *C. elegans*.**'

2.

*The use of proximity labelling to enrich for neuronal proteins in the biochemically challenging model organism is well conceived. However, the criteria for inclusion based on neuronal (or as sometimes-indicated neural proteins) is not clear. Several proteins found in neurons will be widely expressed and shared by other cell types. The basis of this filtering could be better reported. While appreciating the effort from the first attempts to define phospho proteins using panneuronally expressed *pkc-1* up and down regulation I am not sure the challenges of this need reporting per se or at least in the main body of the text.*

Indeed, taking time to provide a better description of the TurboID approach and the justification for statements like the "dramatic change" on UNC-64 would be useful.

We deeply apologize for our oversight. We had missed the description of 'neuron enriched genes'. This is based on the neuron-specific RNA-seq study using FACS (Kaletsky et al., 2016). We added citations to the manuscript.

We think the challenge in using whole body extraction is quite important for other *C. elegans* researchers, as noted in the Discussion. The current major strategy for *C. elegans* phosphoproteomics is to use whole body extraction. However, our data imply that the dataset obtained using whole body extraction do not necessarily represent neural events. This should be noted before explaining the TurboID dataset as the reviewer implied.

Accordingly, we modified **Lines 109-115** and added the reference **(Kaletsky et al., 2016)**. This point is also discussed in Discussion (Lines 297-310).

3.

The ideas around syntaxin are interesting and the model that emerges from the work likely to garner wide interest. It would be useful if the authors could comment on whether mutants harbouring the open form of unc-64, which might occlude the PKC-1 dependent effects, have been tested.

We performed the chemotaxis assay of open syntaxin mutant (unc-64(L166A/E167A)) and examined the genetic interaction with pkc-1(nj3).

Interestingly, unc-64(open) was found to be only slightly affected for the chemotaxis, and could not suppress the low-salt migration phenotype of pkc-1(nj3). This suggests that unc-64(S65A) might have a function other than just switching closed/open forms of syntaxin. We added the result of chemotaxis assay and discussion in Fig. S4C and **Lines 175-177**.

4.

Is there are any confounding (motility phenotypes associated with the key syntaxin mutants used in the study? In particular, the extreme aldicarb insensitivity of the unc-64 (S65A) strain raises this issue. One imagines analysis of sub-behaviours that prelude the arrival during salt chemotaxis would be useful in addressing this

Currently, we do not think the motility defect of unc-64(S65A) affects Chemotaxis Index for the following reasons(i-iii).

(i)

The result of aldicarb sensitivity (Fig. S5A) might have made the impression that unc-64(S65A) has a severe locomotional defect (like 'uncoordinated' phenotype) even in the absence of the drug. Upon revision, we added the locomotion rate (body bending) data in **Fig.S5D**. While unc-64 shows a defect in mobility, the effect is moderate, and is not expected to severely affect the direction of chemotaxis.

(ii)

When conditioned at 50 mM NaCl, WT animals move equally to both directions (Chemotaxis Index, CI \cong 0). In general, when reduced mobility impairs the movement, the CI becomes closer to 0, thus the reduced mobility can not affect CI in 50 mM conditioning, which is already close to zero in wild type. However, unc-64(S65A) mutants showed negative Chemotaxis Index in 50 mM conditioning.

(iii)

unc-64(Ser65) can generate chemotactic bias by acting in the sensory neuron ASER, which is not expected to contribute to locomotion.

5.

The image-based approaches to synaptic release and functional organization are well used. However, some description or reflection of how well these assays map onto the synaptic parameters the authors suggest are key determinants would be useful. It is unclear to this

referee why the recovery from photo bleaching is a measure of vesicle cycling and not confounded by vesicle mobility. As the latter is known to depend on PKC dependent events. If this is an issue, it seems fair that the authors discuss this limitation to their observations.

We are sorry for our insufficient explanation. We are considering that the fluorescence recovery rate reflects the rate of glutamate release, because most of the fluorescence of pHluorin comes from exocytosed membranes. In this study, we photobleached eat-4-fused pHluorin. Since pHluorin inside the vesicle does not absorb photons due to the low pH in the vesicles, it can not be photobleached, thus only pHluorins at the surface of the axon are photobleached. Photobleaching is necessary because at the steady state, rates of exocytosis and endocytosis are balanced and therefore the amount of pHluorin at the axon surface is expected to be constant. Once photobleached, exocytosis rate can be assessed. Our claim is that the rate of exocytosis is affected by pkc-1 and unc-64(S65A), on which increase or decrease in vesicle transportation from other axonal regions may have an indirect effect, but transportation itself does not cause fluorescence recovery.

This methodology using fluorescence recovery is based on previous reports from other labs (See our reference: Samuel et al., 2003).

To avoid misunderstandings, we added explanations in **Line 219-222**.

5.

The authors are to be given credit for using spritz application of glutamate to exposed postsynaptic neuron. I follow their preferred argument here but equally imagine that receptor properties like activation, desensitization and localization will be important in executing the responses that they see.

Particularly given the skilled but limited route of transmitter application.

I think a better description of what is known about the receptor elements that are the focus of their work would be useful.

According to reviewer #1, we added desensitization time constant in Fig S8A, and added some discussion on contribution of desensitization in **Lines 270-275**.

Furthermore, we observed the localization of GLR-1::GFP and AVR-14::mcherry in AIB and added the data in **Fig. S8B**. The fluorescence, especially that of AVR-14::mcherry was dim, and because simultaneous expression of two fluorescent proteins in ASER weakened each fluorescence, we could not observe co-localization of the receptors. However, when separately observed, both GLR-1 and AVR-14 existed throughout the axons and did not localize at a specific site, providing no evidence of differential localization.

6.

I think the shift in response in the avr-14 and glr-1 mutants do evidence the contribution of these receptors. The oocyte expression of single subunits would be more compelling if the authors had evidence that homoligomeric GluCl and AMPA like receptors were consistent with those that are found and used in glutamate transmission at AIB.

We appreciate the reviewer's remark. We tested two glutamate receptors that possibly form heteromeric receptors; AIB expresses two AMPAR subunits: *glr-1* and *glr-2*. Indeed, GLR-1 can form heteromeric receptors with GLR-2 in the interneuron called AVA (See our reference, Mellem et al., 2002). Furthermore, AIB is known to express two inhibitory GluCl receptors, AVR-14 and GLC-4. However, In our previous paper (Sato et al., 2021), we showed that *glc-4* does not contribute to the inhibitory AIB response.

Here, we newly performed in vivo imaging of AIB in the *glr-2* mutant. As a result, *glr-2* exhibited excitatory responses of AIB similar to wild type, which is in contrast to the *glr-1* mutant. Therefore, we concluded homomeric receptors consisting of *glr-1* or *avr-14* functions in AIB for salt chemotaxis. The result is described in **Fig.S7 and Lines 243-248.**

7.

Published work favours the authors' interpretation about relative affinities at these distinct glutamate-binding sites and building this into their discussion would be fair and wise.

We appreciate the reviewer's suggestion.

In this study, it was suggested that there is a difference in the sensitivity and AVR-14 is more sensitive than GLR-1. Previous studies have shown that excitatory and inhibitory glutamate receptors have completely different glutamate binding sites and molecular kinetics. However, it is likely that the sensitivity is unique to each molecule rather than each receptor family: for example, for a molecule related to AVR-14, AVR-15, a glutamate dose-response was measured in *Xenopus* oocyte, similar to the present study. Its EC50 is around 2 mM and is thus much less sensitive compared to AVR-14, indicating that a receptor sensitivity is not necessarily associated to a receptor family.

We added this discussion in **Lines 286-292.**

8.

*Is their previous evidence for robust *glr-1* reconstituted activity?*

We have cited Walker et al., 2006 as a previous evidence for the reconstitution of a GLR-1 receptor. As described in Methods and figure legends, we expressed GLR-1 along with auxiliary subunits SOL-1 and STG-1, following Walker et al.

Minor Comments.

The authors have done a good job at creating a guiding narrative but it feels imprecise in places.

The text from Ln 175 to 183 is difficult to interpret. Is the subject of the text the ref 24 or the author's data in Fig 5A.

Ln 184-184 maybe think about the sense of this.

185-191 describes imaged Ca²⁺ and then suggests that reversal response.

Is this a cellular or behavioural parameter the authors are describing?

We rephrased and reorganized the sentences.

We intended to explain the summary of ref.24 and ref 25 in 175-184.

185-191 just explains the Ca²⁺ response. We observed 'reversal of Ca²⁺ response (from inactivation to activation)', not 'behavioral response called reversal behavior'.

Reviewer #3:

1.

The authors should deposit mass spectrometry raw into the appropriate repositories and processed/human-legible tabular data into supplementary information. For example, quantitative and identity information for all phosphopeptides and unmodified proteins should be included as supplementary material.

Thank you for the suggestion. We uploaded all the raw dataset at ProteomeXchange.

Detail is as the followings:

Project Accession Number: PXD031536

Project Name: Hiroki et al 2022 TurboID Phosphoproteomics RawData

Reviewer Log in:

Username: reviewer_pxd031536@ebi.ac.uk

Password: pVtv5GEz

Furthermore, we added the summary table in the proteomic and phosphoproteomic datasets as **Supplemental Tables**.

2.

For Figs. 2E and 2F, the authors should provide the sequence of the phosphorylated peptides described.

We are sorry for the missing description.

We added the corresponding sequence to **Figs. 2E and 2F**.

3.

*Is there a consensus sequence for PKC-1 substrates? Are the phosphorylated peptides showing high *pkc-1(gf)/pkc-1(nj3lf)* ratios enriched in a consensus sequence motif?*

We analyzed the dataset using a motif discovery algorithm MEME but it did not find any *de novo* consensus motif. Moreover, we did not detect the enrichment of PKC motifs such as RRxSxR.

4.

It would be useful to highlight the point in the volcano plot (Fig. 2D) representing S65 of syntaxin.

Thank you for the suggestion. We added the large purple dot representing UNC-64 phosphopeptide on the plot in **Fig.2D**.

REVIEWERS' COMMENTS

Reviewer #1 (Remarks to the Author):

The authors have addressed all of my comments. Happy to support its publication. Congratulations!

Reviewer #2 (Remarks to the Author):

The authors have attempted to address all comments that I have addressed.

I suggest that given the combined view that clarity about receptor kinetics is key that the data pertaining to receptor activation and desensitization is not relegated to a supplementary. The description of this data is difficult to extract and the time courses described in the figure 8A with a tau value actually supports their preferred model.

I am not convinced that reading through the whole body proteome analysis helps with their narrative but recognize the work involved in this. This would likely be an editorial copy editing decision.

I think the motility phenotype that is relegated to supplementary data should be in main figures.

Thank you to the authors for undertaking a series of additional experiments.

Reviewer #3 (Remarks to the Author):

>> I am pleased to see that the raw data have been deposited to PRIDE and supplementary tables have been included.

2. For Figs. 2E and 2F, the authors should provide the sequence of the phosphorylated

peptides described.

We are sorry for the missing description.

We added the corresponding sequence to Figs. 2E and 2F.

>> These peptide sequences are misleading. I can see from the supplementary table that the peptide sequences include the amino acid that precedes and follows the tryptic peptide in the format like [R].PEPTIDEK.[A]. In Figs. 2E and 2F, this is shown as RPEPTIDEKA for my hypothetical example. This is incorrect and should be fixed.

3.

Is there a consensus sequence for PKC-1 substrates? Are the phosphorylated peptides showing high pkc-1(gf)/pkc-1(nj3lf) ratios enriched in a consensus sequence motif?

We analyzed the dataset using a motif discovery algorithm MEME but it did not find any de novo consensus motif. Moreover, we did not detect the enrichment of PKC motifs such as RRxSxR

>> The lack of enrichment for the RRxSxR motif should be discussed.

2nd Revision Response Letter.

First of all, we deeply thank all the reviewers for helping us.

Point-by point response to the Reviewer Comments.

Reviewer #2

I suggest that given the combined view that clarity about receptor kinetics is key that the data pertaining to receptor activation and desensitization is not relegated to a supplementary. The description of this data is difficult to extract and the time courses described in the figure 8A with a tau value actually supports their preferred model.

We recognize that the feature of receptor desensitization kinetics can be an important factor for those who are familiar with neuroscience or receptors and therefore did add Fig.S8.

However, the key idea of our paper that is most relevant to the mechanism of excitatory/inhibitory switch is the sensitivity of the receptors, as explained in lines 270-273 in Results of the revised manuscript, "This suggests that

under high basal glutamate concentration (0.5-2mM in Fig. 7B, C), the sensitive AVR-14 is already activated and thus desensitized within seconds, while GLR-1 is not activated and thus cannot desensitize at this concentration". We think the description of receptor kinetics in the main figures would somewhat confuse non-specialists who do not have such assumptions. Considering Nature Communications are read by broad range of readers, we think the data should be in Supplemental Figures.

I think the motility phenotype that is relegated to supplementary data should be in main figures.

We thank the reviewer for the suggestion. We moved **Fig. S8D to Fig.3F.**

Reviewer #3

These peptide sequences are misleading. I can see from the supplementary table that the peptide sequences include the amino acid that precedes and follows the tryptic peptide in the format like [R].PEPTIDEK.[A]. In Figs. 2E and 2F, this is shown as RPEPTIDEKA for my hypothetical example. This is incorrect and should be fixed.

We are sorry for the inappropriate expression. we modified the sequence as the reviewer pointed out.

3.

The lack of enrichment for the RRxSxR motif should be discussed.

We thank the reviewer for the suggestion. We added discussion in the **lines 111-113.**